# Tau polarizes an aging transcriptional signature to excitatory neurons and glia

Timothy Wu[1,2], Jennifer M Deger[1,2,3,4], Hui Ye[1,5], Caiwei Guo[1,3,4†], Justin Dhindsa[1,2], Brandon T Pekarek[1], Rami Al-Ouran[1‡], Zhandong Liu[1,6,7], Ismael Al-Ramahi[1,7], Juan Botas[1,7], Joshua M Shulman[1,3,4,5,7]*

[1]Jan and Dan Duncan Neurological Research Institute, Texas Children's Hospital, Houston, United States; [2]Medical Scientist Training Program, Baylor College of Medicine, Houston, United States; [3]Department of Molecular and Human Genetics, Baylor College of Medicine, Houston, United States; [4]Department of Neuroscience, Baylor College of Medicine, Houston, United States; [5]Department of Neurology, Baylor College of Medicine, Houston, United States; [6]Department of Pediatrics, Baylor College of Medicine, Houston, United States; [7]Center for Alzheimer's and Neurodegenerative Diseases, Baylor College of Medicine, Houston, United States

*For correspondence:
joshua.shulman@bcm.edu

Present address: [†]Department of Genetics, Stanford University School of Medicine, Standford, United States; [‡]School of Computing and Informatics, Al Hussein Technical University, Amman, Jordan

Competing interest: The authors declare that no competing interests exist.

**Abstract** Aging is a major risk factor for Alzheimer's disease (AD), and cell-type vulnerability underlies its characteristic clinical manifestations. We have performed longitudinal, single-cell RNA-sequencing in *Drosophila* with pan-neuronal expression of human tau, which forms AD neurofibrillary tangle pathology. Whereas tau- and aging-induced gene expression strongly overlap (93%), they differ in the affected cell types. In contrast to the broad impact of aging, tau-triggered changes are strongly polarized to excitatory neurons and glia. Further, tau can either activate or suppress innate immune gene expression signatures in a cell-type-specific manner. Integration of cellular abundance and gene expression pinpoints nuclear factor kappa B signaling in neurons as a marker for cellular vulnerability. We also highlight the conservation of cell-type-specific transcriptional patterns between *Drosophila* and human postmortem brain tissue. Overall, our results create a resource for dissection of dynamic, age-dependent gene expression changes at cellular resolution in a genetically tractable model of tauopathy.

## Editor's evaluation

Wu et al. have provided a revised manuscript that presents important new findings that start to explain cell type vulnerability and the types of transcriptional changes that occur in the context of neurodegenerative diseases. They cleverly use *Drosophila* for this as they have access to numerous brain cells and exquisite genetic control. They present compelling evidence of transcriptional deregulation and affected pathways in relation to Tau toxicity in a well-controlled study. They also tested if affected pathways modify toxicity but were not successful, however, as pointed out, this can have different reasons. This paper is of broad interest to those in the field of neurodegeneration and neuronal disease and from a methodological point of view to single-cell biologists.

## Introduction

Alzheimer's disease (AD) is a progressive neurodegenerative disorder characterized by extracellular amyloid-beta neuritic plaques and intracellular tau neurofibrillary tangles (*DeTure and Dickson, 2019*; *Scheltens et al., 2021*). Tau neuropathological burden is strongly correlated with cognitive decline, synaptic loss, and neuronal death (*Arriagada et al., 1992*; *Braak and Braak, 1991*; *Gómez-Isla*

*et al., 1997*). Cell-type-specific vulnerability is also an important driver of AD clinical manifestations, including its characteristic amnestic syndrome. Neurofibrillary tangles first appear in the transentorhinal cortex, entorhinal cortex, and CA1 region of the hippocampus, affecting resident pyramidal cells and excitatory glutamatergic neurons; cholinergic neurons of the basal forebrain are also particularly vulnerable (*Mrdjen et al., 2019*; *Fu et al., 2018*). Single-cell RNA-sequencing (scRNAseq) or single-nucleus RNA-sequencing (snRNAseq) are promising approaches to pinpoint cell-type-specific mechanisms in AD, including those that may underlie neuronal vulnerability (*Mathys et al., 2019*; *Grubman et al., 2019*; *Lau et al., 2020*; *Zhou et al., 2020*). Emerging data highlight altered transcriptional states and/or cell proportions for vulnerable versus resilient neurons, including excitatory or inhibitory neurons, respectively (*Leng et al., 2021*). snRNAseq profiles also implicate important roles for non-neuronal cells, including oligodendrocytes, astrocytes, and microglia (*Grubman et al., 2019*; *Lau et al., 2020*; *Zhou et al., 2020*). Microglial expression signatures, including genes with roles in innate immunity, are sharply increased in brains with AD pathology, and an important causal role in AD risk and pathogenesis is reinforced by findings from human genetics (*Bohlen et al., 2019*; *Deczkowska et al., 2018*; *Bellenguez et al., 2022*).

One important limitation to gene expression studies from human postmortem tissue is that only cross-sectional analysis is possible, making it difficult to reconstruct dynamic changes over the full time course of disease. In fact, age is the most important risk factor for AD, which develops over decades (*Masters et al., 2015*; *Villemagne et al., 2013*). Another potential challenge is identifying molecularly specific changes since tau tangle pathology usually co-occurs with amyloid-beta plaques, along with other brain pathologies that can also cause dementia (e.g., Lewy bodies or infarcts) (*Kapasi et al., 2017*). By contrast, animal models permit experimentally controlled manipulations isolating specific triggers and their impact over time. For example, in mouse models of amyloid-beta pathology, scRNAseq and snRNAseq have implicated subpopulations of disease-associated microglia and astrocytes, and similar changes may also characterize brain aging (*Keren-Shaul et al., 2017*; *Habib et al., 2020*). Further, in tau transgenic models, activation of immune signaling by the nuclear factor kappa-light-chain-enhancer of activated B cells (NF-κB) transcription factor within microglia was found to be an important driver of pathological progression (*Wang et al., 2022*). We recently characterized tau- and aging-induced gene expression changes in a *Drosophila melanogaster* tauopathy model, revealing perturbations in many conserved pathways such as innate immune signaling (*Mangleburg et al., 2020*). Over 70% of tau-induced gene expression changes in flies were also observed in normal aging. In this study, we deploy scRNAseq in *Drosophila* to map the cell-specific contributions of age- and tau-driven brain gene expression and identify NFκB signaling as a promising marker of neuronal vulnerability.

## Results

### Single-cell transcriptome profiles of the tau transgenic *Drosophila* brain

Pan-neuronal expression of either wildtype or mutant forms of the human *microtubule-associated protein tau* (*MAPT*) gene in *Drosophila* recapitulates key features of AD and other tauopathies, including misfolded and hyperphosphorylated tau, age-dependent synaptic and neuron loss, and reduced survival (*Wittmann et al., 2001*). We performed scRNAseq of adult fly brains in *tau*$^{R406W}$ transgenic *Drosophila* (*elav>tau*$^{R406W}$) and controls (*elav-GAL4*), including animals aged 1, 10, or 20 days (*Figure 1—figure supplement 1A and B*). The GAL4-UAS expression system is used to express human tau in neurons throughout the central nervous system (CNS) (*Brand and Perrimon, 1993*). The R406W variant in *MAPT* causes frontotemporal dementia with parkinsonism-17, an autosomal-dominant, neurodegenerative disorder with tau pathology (i.e., tauopathy). In flies, wild type and mutant forms of *tau* share conserved neurotoxic mechanisms and cause similar neurodegenerative phenotypes, but *tau*$^{R406W}$ induces a more robust transcriptional response and accelerated course (*Wittmann et al., 2001*; *Bardai et al., 2018*; *Mangleburg et al., 2020*). Following stringent quality control, transcriptome data from 48,111 single cells were available for our initial analyses, including from 6 total conditions (2 genotypes × 3 ages) (*Figure 1—figure supplement 1C and E*). In the integrated dataset, we identified 96 distinct cell clusters grouped by transcriptional signatures, and annotated cell-type identities to 59 clusters using available *Drosophila* brain scRNAseq reference data and established cell markers (*Figure 1A*, *Figure 1—figure supplement 2*, *Figure 1—source data 1*). As expected, most

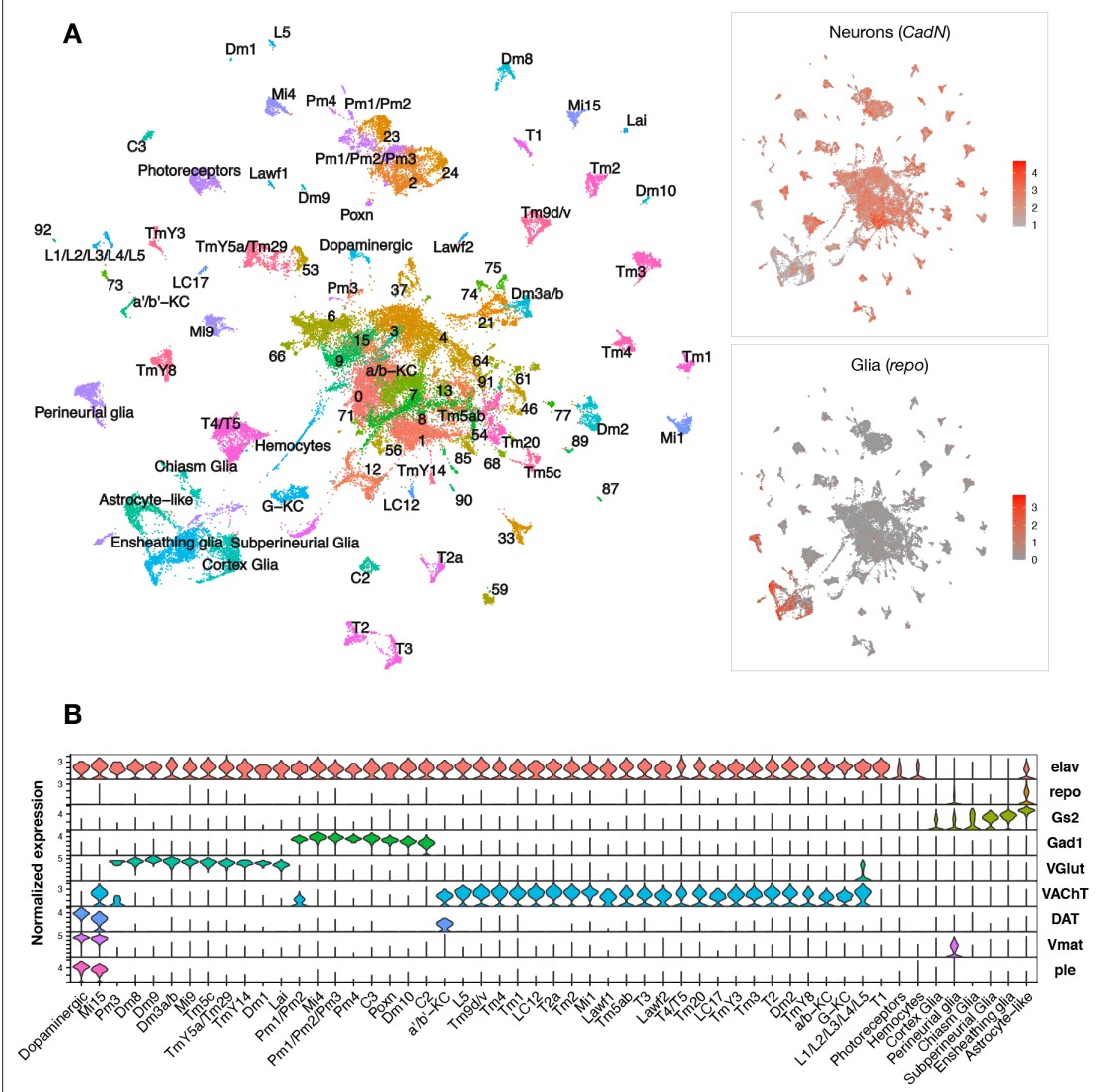

**Figure 1.** Single-cell RNA-sequencing of the adult *Drosophila* brain. (**A**) Uniform manifold approximation and projection (UMAP) plot displays unsupervised clustering of 48,111 cells, including from control (*elav-GAL4/+*) and *elav>tau^R406W* transgenic animals (*elav-GAL4/+; UAS-tau^R406W/+*) at 1, 10, and 20 days. Expression of neuron- and glia-specific marker genes, *CadN* and *repo*, respectively, is also shown. Cell cluster annotations identify heterogeneous optic lobe neuron types, including from the lamina (L1-5, T1, C2/3, Lawf, Lai), medulla (Tm/TmY, Mi, Dm, Pm, T2/3), and lobula (T4/T5, LC). Other identified neuron types include photoreceptors (*ninaC, eya*), dopaminergic neurons (*DAT, Vmat, ple*), and central brain mushroom body Kenyon cells (*ey, Imp, sNFP, trio*). (**B**) Violin plot showing cell-type marker expression across annotated cell clusters. Selected markers include *Elav* (neurons), *repo/Gs2* (glia), *Gad1* (GABA), *VGlut* (glutamate), *VAChT* (acetylcholine), and *DAT/Vmat/ple* (dopamine). See also *Figure 1—figure supplements 1–3* and *Figure 1—source data 1–4*.

The online version of this article includes the following source data and figure supplement(s) for figure 1:

**Source data 1.** *Drosophila* scRNAseq cell cluster annotations.

**Source data 2.** Cell cluster markers.

**Source data 3.** Single-cell RNA-sequencing quality control parameters.

**Source data 4.** *Drosophila* cell-type expression markers.

**Figure supplement 1.** Study design and quality control metrics.

**Figure supplement 2.** Annotating cell identities for 96 cell clusters across 48,111 cells from *Drosophila* brains.

**Figure supplement 3.** Normalized gene expression of general cell-type markers across all defined cell clusters.

cells in the fly brain were neurons (*CadN* expression, n = 42,587), whereas glia were comparatively sparse (*repo* expression, n = 5524). Our dataset comprises a diverse range of cell types. Among all cell clusters, 49% were cholinergic neurons (*VAChT*), 20% were glutamatergic neurons (*VGlut*), 11% were GABAergic neurons (*Gad1*), and 7% were glia (*repo, Gs2*) (*Figure 1B*, *Figure 1—figure supplement 3*). We also identified several major glial subtypes in the fly brain (*Kremer et al., 2017*), including astrocyte-like, cortex, chiasm, subperineurial, perineurial, and ensheathing glia, along with a group of circulating macrophages (hemocytes). Overall, our findings are consistent with results from prior scRNAseq studies of whole adult *Drosophila* brains (*Davie et al., 2018*).

## Tau drives changes in cell proportions in the brain

Leveraging our scRNAseq data and pooling longitudinal samples to permit robust comparisons, we first assessed how tau affects the relative abundance of cell-type subpopulations in the adult brain. We found 16 neuronal and 6 glial clusters with statistically significant changes in cell abundance when comparing tau and controls (*Figure 2A and B*, *Figure 2—source data 1*). Cholinergic mushroom body Kenyon cell neurons in the central complex, which are important in learning and memory, were sharply reduced, likely consistent with developmental toxicity of *tau*, as noted in prior studies of *Drosophila* tauopathy models (*Mershin et al., 2004*; *Kosmidis et al., 2010*). In fact, seven excitatory neuronal clusters, including several cholinergic and glutamatergic cell types, demonstrated significant declines, whereas inhibitory neuronal subpopulations (e.g., Pm and Mi4 GABAergic cells in the visual system) appeared resilient. Conversely, cluster 12 cells appeared more abundant in tau flies; this non-annotated cell type was enriched for neuroendocrine expression markers, *Ms* and *Hug,* as well as a regulator of synaptic plasticity, *Arc1* (*Figure 1—source data 2*). Interestingly, several glial cell types also appeared increased in the brains of tau animals. Ensheathing glia, which showed the largest potential increase, are localized to neuropil in the fly brain and mediate phagocytosis following neuronal injury (*Doherty et al., 2009*; *Freeman, 2015*). In order to confirm these observations, which were based on pooled data across timepoints, we generated additional scRNAseq profiles from 10-day-old *elav>tau$^{R406W}$* and control flies in triplicate samples (69,128 cells; *Figure 2—figure supplement 1*). Overall, 13 out of the 22 significant cell abundance changes were also observed in this replication dataset, including the sharp reduction of excitatory neurons (e.g., Kenyon cells), and the increase in multiple glial clusters (e.g., ensheathing glia) (*Figure 2—figure supplement 1B*, *Figure 2—source data 1*). Non-replicated changes in cell-type abundance may be driven by data from earlier (1 day) or later (20 day) timepoints (*Figure 2B*). Although our experimental design limits cross-sectional analyses at 1 and 20 days, the observed changes in cell abundance were suggestive of a combination of both developmental tau toxicity and progressive, age-dependent neurodegeneration (e.g., neuronal clusters 1, 9, and 12, and astrocyte-like glia). Selected cell-type proportion changes were also recapitulated based on computational deconvolution of available bulk-tissue RNAseq from *tau$^{R406W}$* and control flies at 1, 10, and 20 days by using an independent, published scRNAseq reference dataset (*Figure 2—figure supplement 2*).

Similar to our *Drosophila* tauopathy model, snRNAseq from postmortem human brain tissue has consistently suggested AD-associated increases in glial cell abundance, including astrocytes, oligodendrocytes, microglia, and endothelial cells (*Lau et al., 2020*; *Zhou et al., 2020*). However, one major limitation of both scRNAseq and snRNAseq analysis is that cell-type abundance estimates are relative across the dataset. Therefore, a decline in neuronal subpopulations could lead to inflated abundance estimates of other, stable cell types. Indeed, whereas widespread neuronal loss is highly characteristic of AD (*Davies and Maloney, 1976*; *Braak and Braak, 1991*; *Leng et al., 2021*), systematic histopathological studies in postmortem brain tissue do not support an absolute increase in microglia or astrocyte numbers, but rather a proportional increase in reactive glia in diseased tissues (*Serrano-Pozo et al., 2013*; *Davies et al., 2017*; *Paasila et al., 2019*). We therefore computed confidence intervals for cell abundance changes under an alternative model in which glia were assumed to be unchanging (*Figure 2—figure supplement 3A*). In this more conservative, adjusted analysis, only the neuroendocrine group (cluster 12) was increased and 15 excitatory neuronal subtypes were decreased.

In order to resolve the remaining ambiguity in potential glial cell changes, we performed immunofluorescence on whole-mount *Drosophila* brains (*Figure 2C*). Although the overall intensity of glial nuclear staining (anti-Repo) was increased in *elav>tau$^{R406W}$* flies, quantification revealed no significant

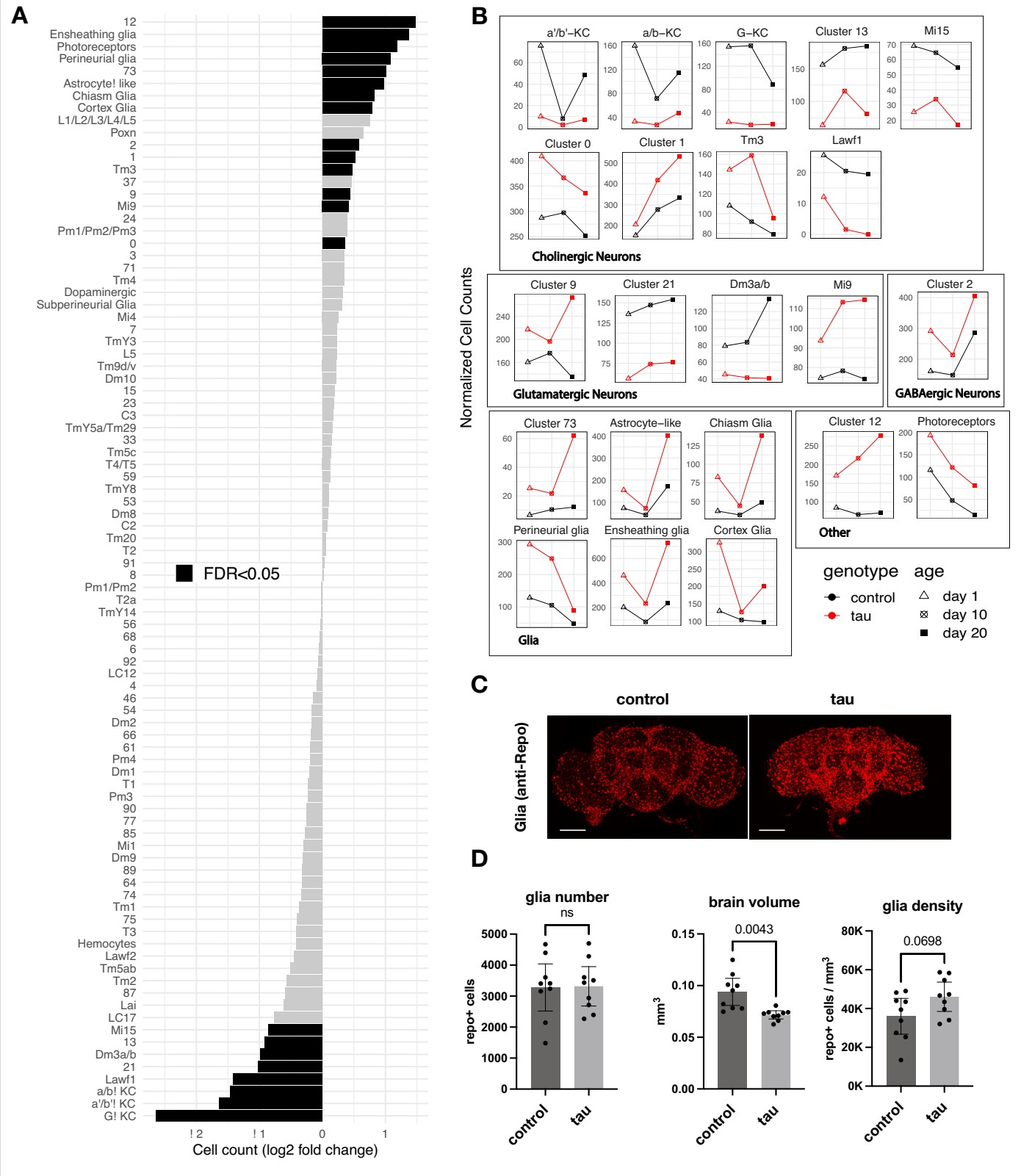

**Figure 2.** Tau-triggered cell proportion changes in the adult brain. (**A**) Log₂-fold change (log2FC) of normalized cell counts between *elav>tau^R406W* (*elav-GAL4/+; UAS-tau^R406W/+*) and control (*elav-GAL4/+*) animals. Timepoints are pooled for each cluster. Cell clusters with statistically significant changes (false discovery rate [FDR] < 0.05) are highlighted in black. Many of these cell abundance changes were replicated in an independent dataset generated from 10-day-old animals (***Figure 2—figure supplement 1***). Since cell-type abundance estimates are relative between clusters, we also performed an

*Figure 2 continued on next page*

*Figure 2 continued*

adjusted analysis in which glia were assumed to be unchanged (**Figure 2—figure supplement 3A**). (**B**) Plots highlight cluster cell counts with significant differences based on pooled timepoint comparisons between *elav>tau^R406W* (red) and control (black) animals, including results for samples collected at 1 day (triangle), 10 days (cross-hatch square), or 20 days (filled square). See **Figure 2—figure supplement 2** for complementary analysis based on deconvolution of bulk brain RNA-sequencing. (**C**) Whole-mount immunofluorescence of adult brains from 10-day-old flies. Glia are stained using the Anti-Repo antibody (red) in control (*elav-GAL4/+*) and *elav>tau^R406W* transgenic flies. Full Z-stack projection is shown. Scale bar = 100 microns. See also **Figure 2—figure supplement 3B** for additional immunostains for nuclei and actin. (**D**) Quantification of glia (Repo-positive puncta), brain volume, and glial density is shown. Statistical analysis employed Welch's T-test with n=9 animals per group and significance threshold p < 0.05. Error bars denote the 95% confidence interval. See also **Figure 2—figure supplements 1–3** and **Figure 2—source data 1**.

The online version of this article includes the following source data and figure supplement(s) for figure 2:

**Source data 1.** Tau-triggered cell proportion changes.

**Figure supplement 1.** Additional scRNAseq from three *tau^R406W* and three control libraries at day 10 post-eclosion.

**Figure supplement 2.** Estimation of cell proportions by deconvolution of bulk-tissue RNA-sequencing.

**Figure supplement 3.** Adjusted tau-triggered cell abundance changes.

---

increase in absolute glial numbers. Instead, we found nominally increased glial density in tau animals after considering their reduced total brain volumes (**Figure 2D**). The increased intensity of antibody staining in tau brains may arise from enhanced antibody penetration since similar changes are also seen for other markers (**Figure 2—figure supplement 3B and C**). Moreover, increased *repo* gene expression was not observed in either scRNAseq or in our previously published bulk-tissue RNAseq (**Mangleburg et al., 2020**). Overall, our results suggest that the apparent increase in glial cell abundance from scRNAseq is likely a consequence of proportional changes in single-cell suspensions due to neuronal loss: in our replication dataset from 10-day-old flies, the proportion of neurons were reduced from 90% to 83% in control versus *elav>tau^R406W* flies. While it is difficult to exclude more modest or selective regional changes, we conclude that similar to human postmortem tissue findings (**Serrano-Pozo et al., 2013**), absolute glial numbers are largely stable following tau expression in the *Drosophila* brain.

## Tau and aging exert cell-specific effects on brain gene expression

To our knowledge, the specific contributions of tau and aging on gene expression across heterogeneous cell types in the adult brain have not been systematically examined. In order to define the impact of aging on brain gene expression, we first quantified cell-specific transcriptional signatures in control flies (*elav-GAL4*) by performing differential expression analyses between the three timepoints from matched cell clusters (**Figure 3A**, **Figure 3—source data 1**). Overall, we define 5998 unique, aging-induced differentially expressed genes. Based on Gene Ontology term enrichment, ribosome/protein translation and energy metabolism pathways were broadly dysregulated during aging, involving the majority of cell types (**Figure 3—source data 2**). We next used linear regression to examine tau-induced differential gene expression within each cell type, including adjustment for age as a covariate. Overall, a total of 5280 unique genes were differentially expressed in at least one or more cell types (**Figure 3B**, **Figure 3—figure supplement 1A**), and these results overlap significantly with our prior bulk RNA-seq in *elav>tau^R406W* flies (**Figure 3—figure supplement 2**). Importantly, 93% of tau-induced differentially expressed genes (n = 4917 out of 5280) were also triggered by aging in control flies (among n = 5998 genes). However, tau and aging appeared to have markedly distinct impacts when considering the distribution of gene perturbations across heterogeneous cell types (**Figure 3C**). Whereas aging broadly perturbed gene expression, tau-triggered changes were sharply polarized to excitatory neurons and glia. Further, the overlap between tau and aging varied across clusters (range = 0–75%) and tau-specific signatures predominated in selected cell types. For example, cholinergic Kenyon cells from the α'/β' mushroom body lobes were among the most vulnerable cell types (**Figure 2A**) and also had the greatest number of tau-induced gene perturbations (**Figure 3B**), which were approximately equally divided between up- and downregulated changes (**Figure 3—figure supplement 1A**, **Figure 3—source data 1**). In fact, among 2289 tau-induced differentially expressed genes within α'/β' Kenyon cells, 2139 (93%) were unique to tau and not similarly triggered in the corresponding cell type in aging control animals. We confirmed that the number of differentially expressed genes and affected cell types does not correspond to the spatial pattern of

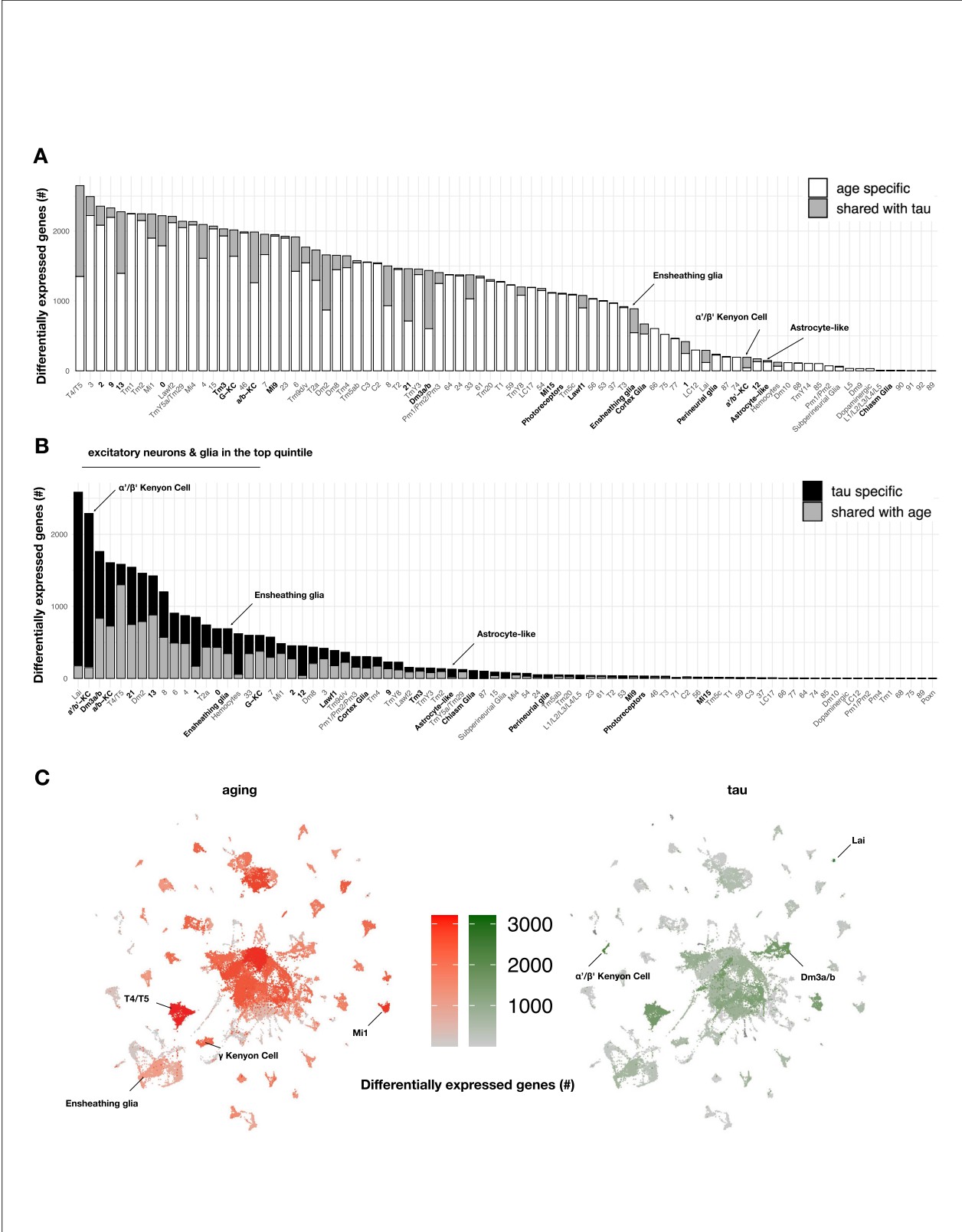

**Figure 3.** Aging- versus tau-triggered brain gene expression changes. (**A**) Aging has widespread transcriptional effects on most brain cell types. Number of aging-induced differentially expressed genes (false discovery rate [FDR] < 0.05) within each cell cluster is shown, based on comparisons of day 1 vs. day 10 and day 10 vs. day 20 in control animals only (*elav-GAL4/+*). For each cell cluster, the number of gene expression changes unique to aging (white) or overlapping with tau-induced changes (gray) is highlighted. Labels for cell clusters with significant tau-induced cell abundance

*Figure 3 continued on next page*

*Figure 3 continued*

changes are shown in bold. (**B**) In contrast with aging, tau induces a more focal transcriptional response, with greater selectivity for excitatory neurons and glia. Number of tau-induced, differentially-expressed genes (FDR < 0.05) within each cell cluster is shown, based on regression models including age as a covariate and considering both control and *elav>tau^{R406W}* animals (*elav-GAL4/+; UAS-tau^{R406W}/+*) at 1, 10, and 20 days. For each cell cluster, the number of gene expression changes unique to tau (black) or overlapping with aging-induced changes (gray) is highlighted. Labels for cell clusters with significant tau-induced cell abundance changes are shown in bold. Tau-induced gene expression changes from single-cell profiles significantly overlap with prior analyses conducted using bulk brain RNA-sequencing (*Figure 3—figure supplement 2*). (**C**) Uniform manifold approximation and projection (UMAP) plots show the number of aging- (red) versus tau- (green) triggered differentially expressed genes within each cell cluster. Color intensity represents the number of differentially expressed genes. See also *Figure 3—figure supplements 1–5* and *Figure 3—source data 1–5*.

The online version of this article includes the following source data and figure supplement(s) for figure 3:

**Source data 1.** Tau- and aging-triggered gene expression changes.

**Source data 2.** Functional pathways from differential expression analysis.

**Source data 3.** Tau-induced gene expression changes in the replication dataset.

**Source data 4.** Cell-type-specific overlaps between tau-induced differentially expressed genes.

**Source data 5.** Cross-sectional tau-induced differential expression.

**Figure supplement 1.** tau-induced differential gene expression analysis and functional enrichment.

**Figure supplement 2.** Overlap between tau-induced adult brain gene expression changes between *Drosophila* scRNAseq and bulk-tissue RNA-sequencing.

**Figure supplement 3.** Volcano plots for selected excitatory neurons and glial populations.

**Figure supplement 4.** Expression of the *MAPT* transgene.

**Figure supplement 5.** Volcano plots for cross-sectional tau-induced differentially expressed genes.

*MAPT* transgene pan-neuronal expression in the brain (*Figure 3—figure supplement 4*); however, it is difficult to exclude the possibility that some vulnerable cell types with high *MAPT* expression might be inadvertently censored from our analyses.

Using functional enrichment analysis, we identify tau transcriptional signatures implicating altered inflammation, oxidative phosphorylation, and ribosomal gene expression (*Figure 3—figure supplement 1B*, *Figure 3—source data 2*). These pathways were prominently disrupted in excitatory neurons of the fly visual system, along with other central brain cholinergic and glutamatergic cell clusters. The pattern of transcriptional perturbation is also consistent with the established susceptibility of the mushroom body and optic lobes to tau-mediated neurodegeneration (*Wittmann et al., 2001*; *Kosmidis et al., 2010*). In other cases, we noted functional enrichments with greater specificity for selected cell clusters, such as altered signatures for mTOR signaling in glutamatergic cluster 21 and Foxo signaling in a subset of neuron types, including lamina intrinsic amacrine (Lai) cells and a cluster receptive to columnar motion (T4/T5). In addition, genes involved in mRNA splicing regulation were perturbed in another group of visual processing cells (T2a) as well as cholinergic cluster 7. Among non-neuronal cells, ensheathing glia, cortex glia, astrocyte-like glia, and hemocytes had the greatest number of tau-driven differential expression changes (*Figure 3—figure supplement 1C*), highlighting signatures related to fatty acid metabolism and synaptic regulation (*Figure 3—source data 2*).

To examine the robustness of our findings, we compared our results on tau-induced, cell-type-specific gene expression changes with the independent dataset from 10-day-old flies. Based on hypergeometric overlap tests of differentially expressed gene sets, expression profiles in two-thirds (61 out of 90) of cell-type clusters from our longitudinal analysis were replicated at 10 days, including several vulnerable excitatory neuron and glial cell clusters (*Figure 3—source data 4*). In secondary analyses, we also analyzed differential expression cross-sectionally, permitting examination of age-dependent changes in specific genes or pathways (*Figure 3—figure supplement 5*, *Figure 3—source data 5*). Overall, when aggregated across all clusters, there was a 90% overlap between the total unique, tau-triggered differentially expressed genes at 10 days between the discovery and replication dataset.

## Tau triggers changes in neuronal innate immune signaling

Whereas most tau-induced genes strongly overlapped with aging, a minority overall were tau-specific (363 out of 5280 gene perturbations). Interestingly, this gene set was significantly enriched for mediators of the innate immune response, particularly NFκB signaling pathway components (*Figure 3—source data 2*). From *Drosophila* bulk brain RNA-seq data, we previously identified seven gene

coexpression modules perturbed by *tau^R406W* expression using weighted correlation network analysis (WGCNA) (*Mangleburg et al., 2020*). Among these, a 236-gene module was strongly enriched for innate immune response genes downstream of NFκB. In our bulk brain RNA-seq data, this module was also activated by wildtype *tau*, but the mutant form, *tau^R406W*, caused a more robust, accelerated response (*Figure 4—figure supplement 1*). In order to better understand the cell type-specific expression patterns, we next examined the innate immune coexpression module in our scRNAseq data. This immune signature was broadly detected in the adult fly brain, including both glia and many neuron types (*Figure 4A*, *Figure 4—figure supplement 2A*). Moreover, expression of the immune module was strongly dysregulated by tau, with 50 out of 90 clusters showing significant changes (*Figure 4B*, *Figure 4—source data 1*). Tau activated the immune signature in the majority of affected cell types (86%, 43 out of 50 clusters). In particular, tau-triggered increases were noted in multiple excitatory neuron clusters (e.g., Dm3 glutamatergic cells in the visual system) as well as non-neuronal cells, including glia (e.g., ensheathing and cortex glia) and hemocytes. Conversely, in a selected subset of seven clusters, tau attenuated expression of the innate immune module (*Figure 4B*), including excitatory neurons in the lamina and several Kenyon cell types that were among the most vulnerable to tau-triggered neuronal loss, based on cell abundance estimates (*Figure 2A*). Other tau-perturbed coexpression modules revealed distinct cell-type-specific patterns (*Figure 4—figure supplement 3*). For example, a module enriched for synaptic regulators was markedly reduced in glia in response to tau, whereas expression was increased in multiple glutamatergic neuron subtypes.

To confirm and extend our analysis of tau- and cell-type-specific gene expression perturbations, we derived a complementary set of 183 transcription factor coexpression networks (regulons) based on our scRNAseq data. Specifically, regulons define coexpressed gene sets in which members are also predicted targets of a specific transcription factor (*Van de Sande et al., 2020*). Overall, clustering cells based on regulon enrichment recapitulates similar, expected relationships between annotated cell types (*Figure 4—figure supplement 4*, *Figure 4—source data 2*), and differential regulon analysis also revealed consistent tau-induced, cell-type-specific transcriptional perturbations (*Figure 4—source data 3*). In particular, we examined the 442-gene regulon comprised of targets of the NFκB transcription factor ortholog in *Drosophila*, Relish (Rel), which is activated downstream of the *Drosophila* Imd (Immune deficiency) pathway, similar to the tumor necrosis factor receptor pathway in mammals (*Myllymäki et al., 2014*). The expression pattern of the Rel regulon and its differential expression in *tau* versus control flies were consistent with our findings for the immune coexpression module derived from bulk RNAseq, which includes both Imd, Rel, and multiple antimicrobial peptides that are activated by Rel (*Figure 4C*). We also obtained consistent results based on a manually-curated, 62-gene set including well-established NFκB signaling pathway members (*Figure 4—figure supplement 2B and C*). Based on our cross-sectional analyses, the pattern of tau-triggered activation of the Rel regulon in selected clusters (e.g., L1-5 lamina neurons and astrocyte-like glia) was age-dependent (*Figure 4—figure supplement 5*). We also experimentally confirmed Rel expression in both neurons and glia in the adult fly brain using an available strain in which the endogenous protein harbors an amino-terminal GFP tag (*Figure 4—figure supplement 2D*).

## Expression signatures for neuronal vulnerability in tauopathy

In order to more directly model the relationship of transcriptional regulation and cellular vulnerability in tauopathy, we integrated regulon expression levels with cell abundance estimates from scRNAseq (*Figure 4—figure supplement 7A*). We hypothesized that innate immune signatures may be predictors of neuronal subtype vulnerability in tauopathy. We implemented regularized multiple regression in which cell-type-specific regulon mean expression served as the predictor variable and tau-triggered cell abundance changes from scRNAseq provided the response variable. The analysis was restricted to cell clusters that show significant declines in *elav>tau^R406W* flies. Out of 183 total regulons, Rel/NFκB activity was prioritized among the top predictors of vulnerability to tau-induced cell loss (*Figure 4D*, *Figure 4—figure supplement 7B*). The Rel regulon remained a robust predictor in an expanded analysis including multiple technical variables as well as expression levels for an additional 2793 curated functional pathways (*Figure 4—source data 4*). Importantly, for this analysis, regulon expression was averaged across both *elav>tau^R406W* and control cells, rather than considering differential expression, and the vulnerable clusters include cell types in which Rel and its targets (Rel regulon) are either activated (e.g., Dm3) or suppressed (e.g., Gamma lobe of the Kenyon cells) in response to tau

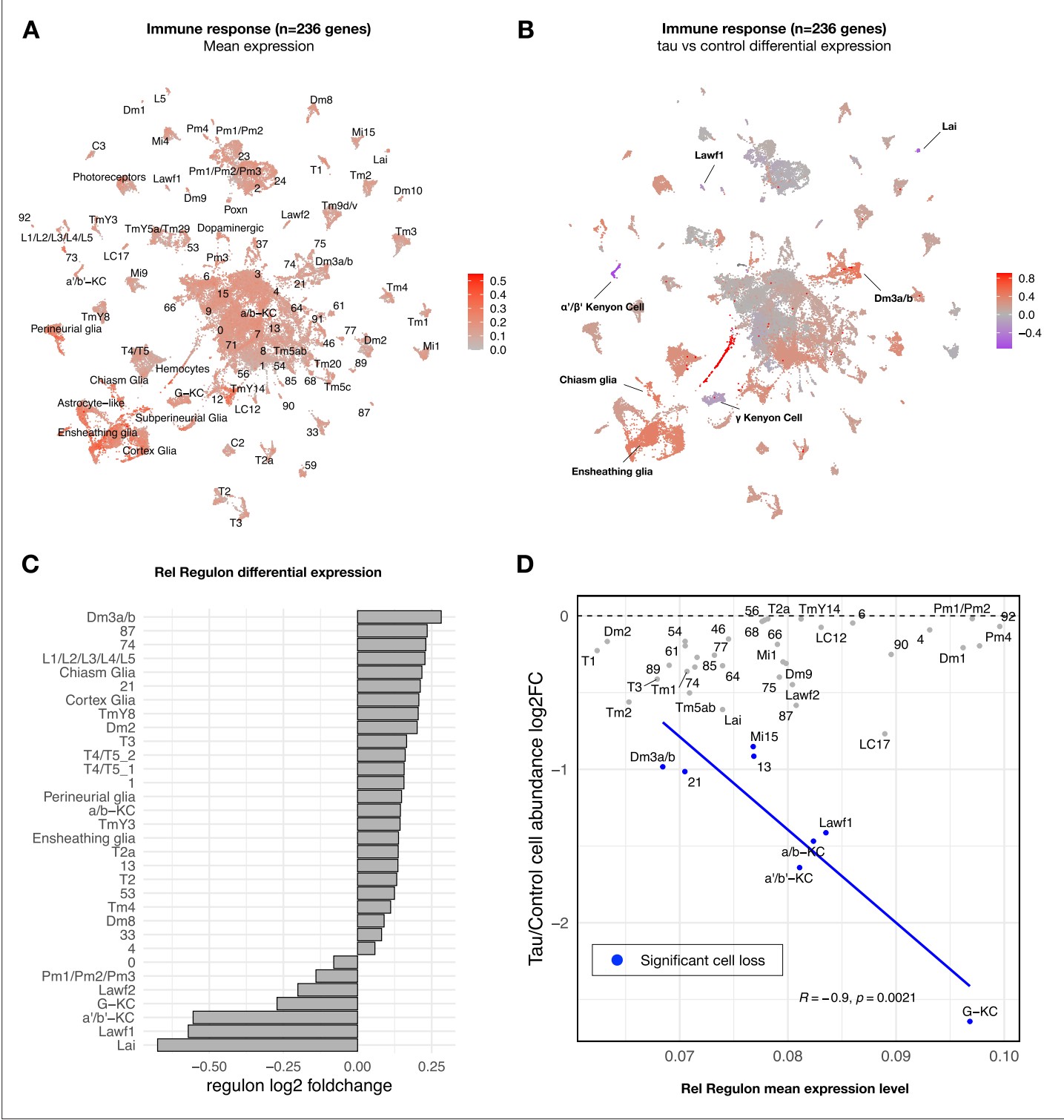

**Figure 4.** Tau-induced changes in innate immune response genes and neuronal vulnerability. (**A**) Innate immune genes are expressed broadly in the adult fly brain, including both neurons and glia. Plot shows mean overall normalized expression by cell cluster among n = 236 genes belonging to a tau-induced coexpression module that is significantly enriched for innate immune response pathways (***Mangleburg et al., 2020***). In this plot, gene expression was averaged across both *elav>tau^R406W* and control cells; similar results are seen when stratifying by either age or genotype (***Figure 4—figure supplement 2A***). See also ***Figure 4—figure supplement 2D*** for experimental confirmation of NF κ B/Rel protein expression in neurons and glia. (**B**) Tau activates or suppresses innate immune response genes in a cell-type-specific manner. Plot shows log₂ fold-change mean expression per cell cluster for the same 236-gene immune response coexpression module, based on comparisons between *elav>tau^R406W* (*elav-GAL4/+; UAS-tau^R406W/+*) and

*Figure 4 continued on next page*

*Figure 4 continued*

control (*elav-GAL4/+*) flies. See also **Figure 4—figure supplement 2B and C** for plots of curated NFκB signaling pathway genes and **Figure 4—figure supplement 3** for similar analyses of other coexpression modules. (**C**) Log2 fold-change in Relish (Rel) regulon gene expression per cluster is shown, based on comparisons between *tau^R406W* and control flies. All results were significant (false discovery rate [FDR] < 0.05) based on regression models including age as a covariate. (**D**) Plot shows overall mean expression of the Rel-regulon (x-axis) versus tau-induced cell abundance change (y-axis). Among clusters with significant, tau-induced cell loss (denoted in blue, FDR < 0.05; see also **Figure 2A**), cell abundance change was inversely correlated with Rel regulon expression (Pearson correlation: $R = -0.9$, p=0.0021). Many other cell types without significant cell abundance changes are also shown in gray. Both control and tau cells are pooled for this analysis. See also **Figure 4—figure supplements 1–8** and **Figure 4—source data 1–4**.

The online version of this article includes the following source data and figure supplement(s) for figure 4:

**Source data 1.** Tau-induced expression changes in innate immune response genes.

**Source data 2.** Regulon coexpression networks.

**Source data 3.** Differential regulon expression analysis.

**Source data 4.** Predictors of tau-triggered cell proportion changes.

**Figure supplement 1.** Mean expression of the innate immune (magenta) module in bulk-tissue RNAseq.

**Figure supplement 2.** Expression of immune response and NFκB genes in *Drosophila* brain.

**Figure supplement 3.** Cell-type-specific expression of tau-induced gene coexpression modules.

**Figure supplement 4.** Unsupervised clustering based on regulon coexpression networks.

**Figure supplement 5.** Cross-sectional differential expression of the Rel regulon (n = 442 genes).

**Figure supplement 6.** Specificity of Rel-GFP animals.

**Figure supplement 7.** Regulons associated with tau-induced cell vulnerability.

**Figure supplement 8.** Pan-neuronal knockdown of *Rel* in *tau^R406W* flies.

(**Figure 4C**). Interestingly, the inverse relationship with cell abundance is recapitulated when restricting consideration of Rel regulon activity in control animals, suggesting that basal NFκB signaling—in the absence of tau—may be a predictive marker for neurodegeneration (**Figure 4—figure supplement 7C**). Specifically, among those cells vulnerable to tau-triggered cell abundance changes, Rel regulon expression is associated with the severity of decline. Besides Rel, the top 3 predictors of vulnerability for tau-induced cell loss include the CrebB and CHES-1 regulons (see **Figure 4—figure supplement 7B** for full list). Interestingly, CrebB—the cAMP response element-binding protein—and its target genes were previously shown to be dysregulated in the *Drosophila* tauopathy model (**Mahoney et al., 2020**), consistent with our finding of CrebB regulon downregulation across many cell types (**Figure 4—source data 3**). In mammals, the conserved CrebB ortholog, CREB, is linked to synaptic plasticity and long-term memory storage, and has also been proposed to interact with the NFκB pathway (**Kaltschmidt et al., 2006**).

In order to directly test whether Rel/NFκB may modify tau-mediated neurodegeneration in a cell-autonomous manner, we used RNA-interference (RNAi) for neuron-specific knockdown of *Rel* and performed histology to detect structural brain degeneration. In these experiments, *elav-GAL4* is used to drive pan-neuronal expression of both *UAS-tau^R406W* and the *UAS-Relish.RNAi* transgenes. However, we did not detect any significant difference in the vacuolar degeneration caused by tau following *Rel* knockdown (**Figure 4—figure supplement 8**). Additional experiments will likely be required definitively resolve the cell-type-specific causal mechanisms (see 'Discussion'); however, our results identify NFκB targets and innate immune signaling as potential markers and/or mediators of vulnerability to tau-mediated neurodegeneration.

## Cross-species overlap of cell-type-specific transcriptional signatures

To establish translational relevance, we next examined the conservation of cell-type-specific transcriptional signatures between *Drosophila* and human brain (**Figure 5A**, **Figure 5—figure supplement 1A**). Using Pearson correlation and considering 5630 conserved genes (1:1 fly/human mapping), we assessed pairwise correspondences between gene expression profiles for all clusters from either our *Drosophila* scRNAseq data (*tau^R406W* + control) and published snRNAseq from human dorsolateral prefrontal cortex (AD cases and control) (**Mathys et al., 2019**). Overall, inferred neuronal and glial cellular identities correlated well across species. Cross-species correlations in cell-type-specific signatures were further replicated in an independent AD case/control snRNAseq dataset from the human

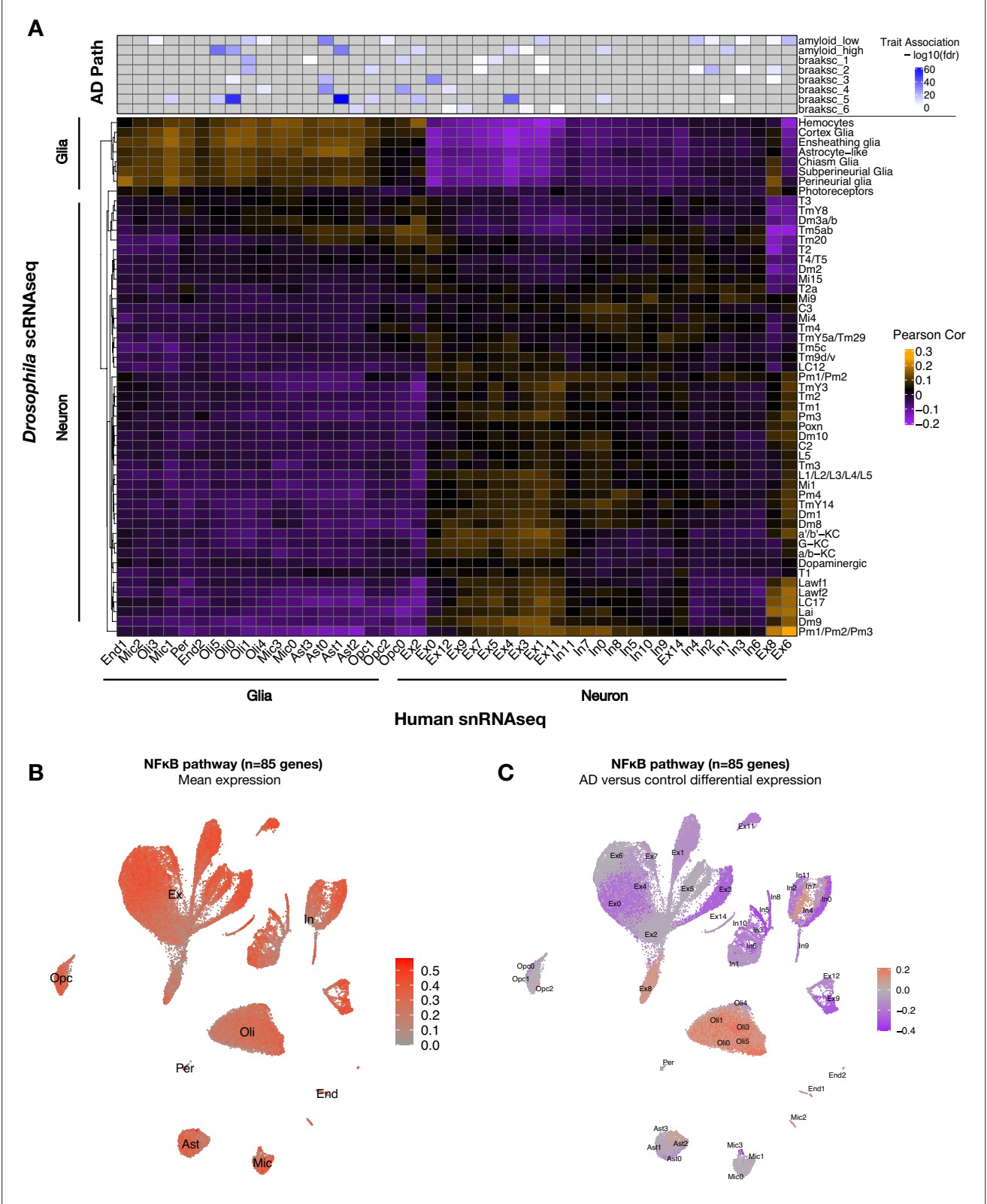

**Figure 5.** Conservation of cell-type-specific gene expression signatures. (**A**) Heatmap shows Pearson correlation of gene expression (5630 conserved, orthologous genes) between annotated cell clusters from *Drosophila* (rows) and human postmortem brain (column). Human brain single-nucleus RNA-sequencing (snRNAseq) was obtained from *Mathys et al., 2019*, including published cell-type associations with amyloid plaque burden and neurofibrillary tangle Braak staging (braaksc) (top). Annotated human cell types include endothelial cells (End), microglia (Mic), oligodendrocytes (Oli),

*Figure 5 continued on next page*

*Figure 5 continued*

pericytes (Per), astrocytes (Ast), oligodendrocyte precursor cells (Opc), excitatory neurons (Ex), and inhibitory neurons (In). (**B**) Innate immune mediators are expressed broadly in the human brain, including in neurons and glia. Plot shows mean expression by cell cluster for 85 human orthologs of NF $\kappa$ B signaling pathway members, based on reprocessing and analysis of the Mathys et al. snRNAseq data. (**C**) Alzheimer's disease (AD) is associated with cell-type-specific perturbation in NF $\kappa$ B signaling genes. Plot shows log$_2$ fold-change mean expression per cell cluster for the same 85 NF $\kappa$ B signaling genes, based on comparisons of brains with AD pathology versus controls. See also *Figure 5—figure supplements 1–3* and *Figure 5—source data 1*.

The online version of this article includes the following source data and figure supplement(s) for figure 5:

**Source data 1.** Cell-type-specific, Alzheimer's disease (AD)-associated gene expression changes from human brain.

**Figure supplement 1.** Cross-species gene expression correlation of all *Drosophila* cell clusters in this study.

**Figure supplement 2.** Replication of gene expression correlation between *Drosophila* scRNAseq and snRNAseq from control human subjects.

**Figure supplement 3.** Cell-specific Rel/NF $\kappa$ B regulon differential expression in *MAPT*$^{P301L}$ transgenic mice.

entorhinal cortex (*Grubman et al., 2019*; *Figure 5—figure supplement 1B*). Similar results were also obtained in a complementary analysis leveraging a published *Drosophila* scRNAseq dataset (wildtype flies only) and excluding human brains with AD pathology (controls only) (*Figure 5—figure supplement 2*). The resulting correlation map can enable integrative, cross-species analyses. For example, a human microglial subcluster (Mic1) notable for association with high tau neuropathological burden was correlated with the ensheathing glia cluster from *Drosophila*, indicating shared characteristic transcriptional signatures (*Figure 5A*). Moreover, these two cell types showed significantly overlapping gene expression changes in association with AD pathology (human brain) or following pan-neuronal expression of *tau*$^{R406W}$ (*Drosophila*) (hypergeometric test, p=4.83 × 10$^{-5}$) (*Figure 5—source data 1*). This result suggests that tau pathology may indeed be an important driver of Mic1 transcriptional changes in disease.

As introduced above, mediators of innate immunity are also highly conserved across species. Similar to *elav>tau*$^{R406W}$ flies, we confirmed consistent NFκB pathway expression in excitatory neurons and microglia in transgenic mice harboring a *MAPT*$^{P301S}$ transgene (*Lee et al., 2021*; *Figure 5—figure supplement 3*). Next, leveraging the Mathys et al. human snRNAseq data, we confirmed that NFκB signaling pathway genes are expressed across most cell types in human postmortem brain tissue, including both neurons and glia (*Figure 5B*). In the context of AD pathology, NFκB pathway gene expression appeared strongly downregulated in most neurons from the dorsolateral prefrontal cortex, which are highly susceptible to degeneration, whereas expression was increased among oligodendrocytes, microglia, and astrocytes (*Figure 5C*). Interestingly, a subset of excitatory and inhibitory neuronal subclusters (Ex8 and In4, respectively) showed an AD-associated increase in expression. Thus, human brains with AD pathology are also characterized by widespread changes in NFκB innate immune signaling, including either activation or attenuation in many distinct neuronal and non-neuronal subtypes.

## Discussion

Aging is the most important risk factor for AD, influencing both disease onset and progression. Based on longitudinal, single-cell analysis in *Drosophila*, we discover that tau and aging activate strongly overlapping transcriptional responses: 93% of tau-induced differentially expressed genes are also perturbed by aging in control animals. Instead, tau and aging are distinguished by their spatial and cell-type-specific impacts. Aging has a global influence on brain gene expression, affecting most brain cell types. By contrast, tau has a focal impact, polarizing the transcriptional response to a handful of cell types, including excitatory neurons and glia. The strong overlap between tau- and aging-induced gene expression signatures agrees with our prior analyses of bulk brain tissue (*Mangleburg et al., 2020*). We and others have also documented similar findings in AD mouse models, including both *MAPT* and *amyloid precursor protein* transgenics (*Wan et al., 2020*; *Cummings et al., 2015*; *Gjoneska et al., 2015*; *Matarin et al., 2015*; *Hargis and Blalock, 2017*). By contrast with animal models, cross-sectional studies of human postmortem tissue make it difficult to disambiguate the impact of aging from disease pathology on the brain transcriptome. However, our cross-species analyses highlight that most human brain cell types share transcriptional signatures with counterparts in the *Drosophila* brain.

These correspondences comprise a cross-species atlas enabling studies of controlled experimental manipulations (e.g., tau vs. aging) on homologous cell clusters between humans and flies.

Mechanistic dissection of cell-type-specific vulnerability promises to reveal drivers for the earliest clinical manifestations of AD, such as the characteristic memory impairment accompanying the loss of excitatory neurons in hippocampus and associated limbic regions (*Mrdjen et al., 2019*; *Fu et al., 2018*). Given the transcriptional overlaps, one attractive model is that aging establishes a spatial pattern of vulnerable cell states that templates the subsequent tau-triggered neurodegeneration. However, as noted above, aging has wide-ranging impact across the brain and many cell types with robust aging-induced transcriptional responses in *Drosophila* are, in fact, resilient to tau-mediated neurodegeneration based on cell proportion changes (e.g*.*, clusters 2, 3, and T4/T5; *Figures 2A and 3A*). Moreover, the overlap between tau and aging does not reliably predict those cell types that are most vulnerable to neuronal loss (*Figure 3A and B*). Differentially expressed genes triggered by tau and aging are nevertheless similarly enriched for many common biological pathways that may provide clues to cell-type-specific mechanisms of vulnerability in neurodegeneration. Specifically, we document shared expression signatures for altered synaptic regulation, protein translation, lipid metabolism, and oxidative phosphorylation across heterogeneous cell populations, including excitatory neuron types that are particularly vulnerable to tau. Similar pathways have been implicated based on snRNAseq analyses from human postmortem brain (*Mathys et al., 2019*; *Grubman et al., 2019*; *Lau et al., 2020*) and several mouse AD models, including *MAPT* transgenics (*Wang et al., 2022*; *Lee et al., 2021*; *Habib et al., 2020*; *Zhou et al., 2020*).

Among the many dysregulated molecular processes, aging is characterized by a systemic pro-inflammatory state that has been called 'immunosenescence' or 'inflamma-aging' (*Shaw et al., 2013*; *Hou et al., 2019*). Genes encoding regulators of immunity, including *TREM2, CR1*, and many others, have been strongly implicated in AD susceptibility by human genetics (*Bellenguez et al., 2022*), and abundant evidence now supports a key role for many such genes among glial cells (*Wang et al., 2015*; *Zhou et al., 2020*; *Keren-Shaul et al., 2017*). We previously identified an age-associated *Drosophila* innate immune response signature that is amplified by tau (*Mangleburg et al., 2020*). Here, we significantly extend these observations, leveraging the cellular resolution afforded by single-cell profiles. First, we discover that this immune coexpression module, including many NFκB/Rel signaling factors and targets, is broadly expressed in the adult fly brain, including both neurons and glia, and we confirm similar findings in snRNAseq data from human postmortem brain. Second, we show that tau can either activate or attenuate NFκB immune pathways in a cell-type-specific manner, with tau-triggered decreases in expression apparent in neurons with the greatest proportional cell loss. Lastly, models integrating cell-type-specific gene transcriptional expression and cell abundance changes suggest that basal Imd signaling strength (i.e., Rel regulon activity) predicts the severity of tau-triggered neuronal decline among susceptible cell types. Overall, our results suggest that besides the well-established requirements in glia (see below), innate immune response pathways may also have important, cell-autonomous roles in modulating neuronal vulnerability to tau pathology in AD. Indeed, both insect and mammalian neurons express evolutionary-conserved innate immune signaling pathways, including Toll-like receptors and NFκB signal transduction components, and these pathways can be triggered by infection or other cellular insults (*Lehnardt et al., 2003*; *Tang et al., 2007*; *Cao et al., 2013*; *Cho et al., 2013*; *Petersen et al., 2013*; *Welch et al., 2022*). In addition, NFκB immune signaling pathways have been coopted for diverse, non-canonical functions, such as in neurodevelopment and synaptic plasticity (*Okun et al., 2011*; *Gutierrez and Davies, 2011*; *Nguyen et al., 2020*). Knockdown of *Rel* in *Drosophila* neurons has previously been shown to promote survival in non-transgenic, wildtype animals (*Kounatidis et al., 2017*), whereas activation of the Rel signaling pathway leads to neurodegeneration (*Cao et al., 2013*). In addition, a recent reanalysis of snRNAseq data from *Mathys et al., 2019* revealed AD-associated perturbation of NFκB immune pathways in excitatory neurons, possibly triggered by DNA double-strand breaks (*Welch et al., 2022*). Although experimental manipulation of *Rel* in the *elav>tau^{R406W}* model did not alter tau-mediated neurodegeneration, additional studies may be required to definitively resolve the potential cell-type-specific causal contribution(s) of NFκB/Relish. Our negative result could reflect poor sensitivity and variability of the histologic assay or it may be necessary to use alternate neuronal drivers restricted to the adult brain.

Our scRNAseq analyses also highlight a robust, tau-induced transcriptional response among *Drosophila* glia. This result is consistent with several brain gene expression studies from both humans and mouse models that strongly implicate altered transcriptional states and/or increased numbers of AD-associated glial subtypes, including oligodendrocytes, astrocytes, and microglia (*Mathys et al., 2019*; *Grubman et al., 2019*; *Lau et al., 2020*; *Zhou et al., 2020*; *Habib et al., 2020*). Although our analyses initially suggested a possible tau-triggered increase in glial abundance in the brain, on direct examination, we documented stable absolute numbers but increased density of glia due to brain atrophy. Systematic histopathological studies in human brain tissue have similarly revealed predominantly reactive changes with overall stable numbers of both astrocytes and microglia (*Serrano-Pozo et al., 2013*). We conclude that potential increases in disease-associated glia inferred exclusively from single-cell profiles should be interpreted cautiously, and additional experimental investigations may ultimately be required to resolve whether they result from (i) absolute changes in cell number, (ii) activation and/or transformation of cell states, or (iii) proportional changes due to primary perturbations in other brain cell types. Nevertheless, glial-specific experimental manipulations of immune regulators in both *Drosophila* and mammalian models, including NFκB signaling (flies and mice) and the AD susceptibility gene *TREM2* (mice), can potently modify neurodegeneration, consistent with cell non-autonomous requirements (*Walter, 2016*; *Kounatidis et al., 2017*; *Petersen et al., 2012*; *Hakim-Mishnaevski et al., 2019*; *Fuhrmann et al., 2010*; *Town et al., 2008*; *Leyns and Holtzman, 2017*; *Wang et al., 2022*). By contrast with mammals, glia represent only 5–10% of all cells in the *Drosophila* brain (*Ito et al., 1995*; *Schmidt et al., 1997*; *Awasaki et al., 2008*). Nevertheless, *Drosophila* glial subtypes recapitulate the diversity of functions and morphologies of mammalian glia (*Doherty et al., 2009*; *Freeman, 2015*; *Kremer et al., 2017*; *Stork et al., 2012*). Although the myeloid hematopoietic lineage is not present in flies, which therefore lack microglia, ensheathing glia can similarly respond to cellular injury and scavenge debris (*Doherty et al., 2009*). Indeed, our cross-species analysis demonstrates shared transcriptional signatures between corresponding glial subtypes, consistent with our findings of conserved responses to tau-mediated neuronal injury. In future work, it will be interesting to further dissect both the cell-autonomous and non-cell-autonomous drivers underlying both the neuronal and glial responses to tauopathy.

The *elav>tau^{R406W}* flies selected for this study share conserved downstream mechanisms of neurotoxicity with wildtype tau (*Bardai et al., 2018*) and have been widely used as an experimental model for investigations of both AD and other tauopathies, including frontotemporal dementia. Nevertheless, one potential caveat is the absence of amyloid-beta peptide, which is also an important driver of gene expression changes in AD, including innate immune transcriptional signatures (*Keren-Shaul et al., 2017*; *Wan et al., 2020*). Another potential limitation is that the *elav-GAL4* driver activates *tau* expression during developmental stages, and the observed changes in cell-abundance or gene expression may therefore reflect this time course. For example, tau developmental toxicity has been shown to cause malformation of mushroom body structures (*Kosmidis et al., 2010*), and this phenotype likely explains the reductions in several cell clusters in our dataset. While our study was under review, a complementary, single-cell transcriptome analysis using the *nsyb>tau^{P301L}* model was published, in which transgene expression is expected to be more restricted within the adult brain (*Praschberger et al., 2023*). While there were overlaps in the vulnerable cell types for both the *elav>tau* and *nsyb>tau* models (e.g., excitatory cholinergic neuron subtypes, like γ-KC, α'/β'-KC, and T4/5), there were also some notable distinctions—the inhibitory C2 cell cluster, which is GABAergic, was highlighted only in the *nsyb* model. Further comparisons are somewhat limited by other experimental and analytic design differences between the studies. Nevertheless, the *elav>tau^{R406W}* model is well established to recapitulate aging-dependent, neuronal loss and progressive CNS dysfunction (*Wittmann et al., 2001*). Indeed, our longitudinal design reveals suggestive age-dependent cell abundance changes among several cell types (e.g., clusters 1, 9, and 12, along with astrocyte-like glia), and cross-sectional analyses also reveal evidence for progressive transcriptional changes. It will be important to perform additional studies, perhaps using inducible driver systems, to more systematically dissect the dynamic time course of tau neurotoxic mechanisms, including differentiating developmental versus degenerative changes that accompany brain aging.

## Materials and methods

**Key resources table**

| Reagent type (species) or resource | Designation | Source or reference | Identifiers | Additional information |
|---|---|---|---|---|
| Antibody | Rabbit polyclonal anti-GFP | GeneTex | Cat#GTX113617; RRID:AB_1950371 | IF(1:500) |
| Antibody | Alexa 647 goat polyclonal anti-rabbit IgG (H+L) | Jackson ImmunoResearch | Cat#111-605-003 | IF(1:500) |
| Antibody | CyTM3 AffiniPure goat polyclonal anti-mouse (H+L) | Jackson ImmunoResearch | Cat#115-165-003 | IF(1:500) |
| Antibody | Alexa Fluor 488 donkey polyclonal anti-mouse IgG (H+L) | Jackson ImmunoResearch | Cat#715-545-150 | IF(1:500) |
| Antibody | Cy3TM3 AffiniPure goat polyclonal anti-rat IgG (H+L) | Jackson ImmunoResearch | Cat#112-165-003 | IF(I:500) |
| Antibody | Mouse monoclonal anti-repo | DSHB | Cat#8D12 | IF(1:500) – glial counting IF(1:50) – Rel costain |
| Antibody | Rat monoclonal anti-Elav | DSHB | Cat#7E8A10; RRID:AB_528218 | IF(1:100) |
| Antibody | Mouse monoclonal anti-Rel | DSHB | Cat#21F3; RRID:AB_1553772 | IF(1:500) |
| Chemical compound, reagent | Conjugated A488-Phalloidin | Thermo Fisher | Cat#A12379 | IF(1:500) |
| Chemical compound, drug | Dispase | Sigma-Aldrich | Cat#D4818; | |
| Chemical compound, drug | Collagenase I | Invitrogen | Cat#17100-100 | |
| Chemical compound, drug | NucBlue and Propidium iodide | Invitrogen | Cat#R37610 | |
| Chemical compound, drug | Vectashield antifade mounting medium | Vector Laboratories | Cat#H-1000-10 | |
| Commercial assay or kit | Chromium Single Cell Gene Expression 3' v3.1 | 10x Genomics | Cat#PN-1000268 | |
| Genetic reagent (*Drosophila melanogaster*) | *elav$^{C155}$-GAL4* | Bloomington *Drosophila* Stock Center | BDSC:458 | |
| Genetic reagent (*D. melanogaster*) | *w$^{1118}$; UAS-Tau$^{R406W}$* | Lab: Dr. Mel B. Feany, PMID:11408621 | N/A | ***Wittmann et al., 2001*** |
| Genetic reagent (*D. melanogaster*) | *Rel-GFP* | Bloomington *Drosophila* Stock Center | BDSC:81268 | *y$^1$ w\*; PBac{GFP.FPTB-Rel}VK00037* |
| Genetic reagent (*D. melanogaster*) | *UAS-Rel.RNAi-2* | Bloomington *Drosophila* Stock Center | BDSC:33661 | *y$^1$; P{TRiP.HMS00070}attP2* |
| Genetic reagent (*D. melanogaster*) | *UAS-Rel.RNAi-1* | Vienna *Drosophila* Resource Center | VDRC:49414 | *P{GD1199}v49414* |
| Software, algorithm | Imaris Microscopy Image Analysis Software 9.9.1 | https://imaris.oxinst.com/ | | Oxford Instruments |
| Software, algorithm | Prism 9.4.1 | https://www.graphpad.com/scientific-software/prism/ | | GraphPad |
| Software, algorithm | ImageJ | https://imagej.nih.gov/ij/ | | NIH |
| Software, algorithm | Cell Ranger 4.0.0 | https://support.10xgenomics.com/single-cell-gene-expression/software/pipelines/latest/what-is-cell-ranger | | 10x Genomics |

| Reagent type (species) or resource | Designation | Source or reference | Identifiers | Additional information |
|---|---|---|---|---|
| Software, algorithm | Seurat v3 | https://doi.org/10.1016/j.cell.2019.05.031 | | *Stuart et al., 2019* |
| Software, algorithm | DoubletFinder 2.0.3 | https://github.com/chris-mcginnis-ucsf/DoubletFinder | | *McGinnis et al., 2019* |
| Software, algorithm | Scmap 1.9.3 | https://bioconductor.org/packages/release/bioc/html/scmap.html | | *Kiselev et al., 2018* |
| Software, algorithm | Optic lobe neural network classifier | https://static-content.springer.com/esm/art%3A10.1038%2Fs41586-020-2879-3/MediaObjects/41586_2020_2879_MOESM7_ESM.zip | | *Özel et al., 2021*, Supplementary Data Appendix 1, Python/R code |
| Software, algorithm | pySCENIC 0.12.0 | https://github.com/aertslab/pySCENIC | | *Van de Sande et al., 2020* |
| Software, algorithm | DESeq2 1.34.0 | https://bioconductor.org/packages/release/bioc/html/DESeq2.html | | *Love et al., 2014* |
| Software, algorithm | MuSiC 0.1.1 | https://github.com/xuranw/MuSiC | | *Wang et al., 2019* |
| Software, algorithm | MAST 1.20.0 | https://bioconductor.org/packages/release/bioc/html/MAST.html | | *Finak et al., 2015* |
| Software, algorithm | WEBGESTALTR 0.4.4 | https://github.com/bzhanglab/WebGestaltR | | *Wang et al., 2013* |
| Software, algorithm | Glmnet 4.1-4 | https://cran.r-project.org/web/packages/glmnet/index.html | | *Friedman et al., 2010* |
| Software, algorithm | Caret 6.0-92 | https://cran.r-project.org/web/packages/caret/index.html | | *Kuhn, 2008* |
| Software, algorithm | DRSC Integrated Ortholog Prediction Tool (DIOPT) | https://www.flyrnai.org/diopt | | *Hu et al., 2011* |
| Software, algorithm | gProfiler2 0.2.1 | https://cran.r-project.org/web/packages/gprofiler2/index.html | | *Raudvere et al., 2019* |
| Software, algorithm | SCTransform 0.3.3 | https://github.com/satijalab/sctransform | | *Stuart et al., 2019* |

## Human subjects

No new data from human subjects were generated for this study. Previously published, available snRNAseq data from human postmortem brain were obtained from *Mathys et al., 2019* and *Grubman et al., 2019* in order to evaluate cross-species correspondences in cell-type-specific expression signatures. The Mathys data is comprised of snRNAseq from the dorsolateral prefrontal cortex (DLPFC) from 48 brain autopsies with varying AD neuropathology (amyloid plaque and tau neurofibrillary tangle burden), including 24 with no significant pathology (controls) and 24 cases with mild to severe AD pathology. Subjects were balanced for sex (12 males and 12 females), and age (median age at death = 87 for both groups). The Grubman data is comprised of snRNAseq from the entorhinal cortex of 12 brain autopsies, including 6 AD pathological cases and 6 controls without significant AD pathology. Subjects in the Grubman data were also age-matched, with a median age of 83 and 80 for the AD case and control groups, respectively.

## *Drosophila* stocks and husbandry

For scRNAseq libraries generated in this study, *w[1118]; UAS-tau[R406W]* flies (0N4R isoform, 383 amino acids), described in *Wittmann et al., 2001*; *Mangleburg et al., 2020* were crossed with the pan-neuronal driver *elav[C155]-Gal4*, producing the experimental genotypes: *elav-Gal4/+;UAS-tau[R406W]/+* or *elav-Gal4/Y; UAS-tau[R406W]/+*. In order to minimize genetic background as a potential confounder,

UAS-tau[R406W] strains used in this study were backcrossed with w[1118] for five generations as previously described (*Guo et al., 2018*). Controls were generated by outcrossing *elav-Gal4* with w[1118] animals, producing *elav-Gal4/+* or *elav-Gal4/Y*. Adult progeny from experimental crosses were subsequently aged to 1, 10, or 20 d for dissection and library generation. Flies were raised on standard molasses-based media at 25°C in ambient lighting. We also utilized a *Rel-GFP* strain (*y, w; PBac{GFP.FPTB-Rel} VK00037*), which is an endogenous protein trap allele, encoding a fusion protein with GFP at the Rel amino-terminus. For the histology experiments, *elav-Gal4/Y;UAS-tau[R406W]/+* animals were crossed with *UAS-Rel.RNAi-1* (VDRC: v49414), *UAS-Rel.RNAi-2* (TRiP: HMS00070), or w[1118]. Resulting female progeny with both the *UAS-tau[R406W]* transgene and RNAi (or controls) were aged to 10 d and prepared for histology.

## *Drosophila* brain histology

*Drosophila* heads were fixed in 8% glutaraldehyde (Electron Microscopy Sciences) at 4°C for 10 d, followed by paraffin embedding and microtome sectioning as previously described in *Chouhan et al., 2016*. Serial 5-µm-thick coronal sections were prepared for the whole head, mounted onto microscopy slides, and stained with hematoxylin and eosin. Bright-field microscopy images were acquired using the Leica DM 6000B system. For quantification, the number of vacuoles greater than 5 um in diameter in an ~50 um stack comprising of the ellipsoid body, fan-shaped body, and posterior commissure. The mean number of vacuoles per section was computed per animal. Statistical testing between conditions was performed using Welch's *t*-test.

## *Drosophila* brain dissociation

For scRNAseq profiling of *elav >tau[R406W]* and control flies, 16–18 dissected and intact *Drosophila* brains were combined and dissociated for each experimental condition (six total samples: 2 genotypes × 3 timepoints). An equal number of male and female animals were combined for each condition. For the replication dataset, triplicate samples (biological replicates) for the identical *elav>tau[R406W]* and control genotypes were prepared at day 10 (six total samples). Adult fly brains were dissected out of the cuticle using sharp forceps in 1X PBS and dissociated following published protocols (*Davie et al., 2018*). Dissected brains in solution were first centrifuged at 800 × *g* for 3 min, resuspended, and dissociated by incubating with 50 uL of dispase (3 mg/mL, Sigma) and 75 uL of collagenase I (100 mg/mL, Invitrogen) for 2 hr at 25°C while shaking at 500 RPM. Cell suspensions were mixed by gentle pipetting 3–4 times every 5 min in the first hour, and every 10 min in the second hour. Resulting cell suspensions were pelleted by centrifugation at 400 × *g* for 5 min at 4°C, washed in 1000 uL ice-cold PBS, pelleted, and resuspended in 400 uL ice-cold PBS with 0.04% bovine serum albumin. Cell suspensions were passed through a 10 um pluriStrainer cell strainer (pluriSelect) to ensure that undissociated tissue were removed and a single-cell suspension was obtained. Cell concentration and viability were assessed using a hemocytometer under a fluorescent microscope after staining with NucBlue and Propidium iodide (Invitrogen). Fresh, intact single-cell suspensions were immediately used for single-cell library preparation.

## Single-cell library preparation and sequencing

Single-cell libraries were prepared per the manufacturer's protocol for the Chromium Single Cell Gene Expression 3' v3.1 kit (10x Genomics) by the BCM Single Cell Genomics Core. 16,000 cells were added to each channel with a target recovery rate of 10,000 cells per library. Cells, reverse transcription (RT) reagents, gel beads containing barcoded oligonucleotides, and oil were loaded on a Chromium controller (10x Genomics) to generate single-cell Gel Bead-In-Emulsions (GEMs) where full-length cDNA was synthesized and barcoded for each individual cell. GEMs were subsequently broken and cDNAs from each single cell were pooled. Following clean up using Dynabeads MyOne Silane Beads (Invitrogen), cDNA was amplified by PCR. The amplified product was fragmented to optimal size before end-repair, A-tailing, and adaptor ligation. Final library was generated by amplification. Completed libraries were sequenced using the Baylor Genomic and RNA Profiling Core on the Illumina NovaSeq 6000 platform with a minimum depth of 300,000,000 reads per sample (on average 463 M reads per sample). A total of 12 high-quality libraries were generated (six libraries for the discovery and replication datasets, respectively). Illumina BCL files were demultiplexed into FASTQ files by calling the Cell Ranger 4.0.0 *mkfastq* function. FASTQ files were aligned to the *Drosophila* reference genome

(BDGP6.22.98) and quantified using the Cell Ranger 4.0.0 *count* pipeline. The human microtubule-associated protein tau (MAPT) mRNA coding sequence (CDS) (isoform 3, NCBI Reference Sequence NM_016834.5:151–1302) along with a short SV40 3'UTR sequence was appended to the *Drosophila* reference genome for assessing MAPT transgene expression levels. Given the 10× recovery rate estimations, the cell calling algorithm in Cell Ranger was applied by setting the `--expect-cells` parameter in *count* to 10,000 for each library, thus filtering out partitions that likely did not contain single cells. Cell ranger alignment metrics for each library are available in *Figure 1—source data 3*. Filtered count matrices were loaded into Seurat v3 in R for additional quality control and downstream analyses. Cells were removed from the data object if the number of unique genes per cell were less than 200 or greater than 3000, or if the proportion of mitochondrial reads per cell was greater than 20%. Filtered count matrices from Cell Ranger are available to download with the *Drosophila* scRNAseq data on the Synapse AMP-AD Knowledge Portal.

## Normalization, integration, and clustering

Gene expression was first normalized independently per library using a regularized negative binomial regression approach as implemented by SCTransform (*Stuart et al., 2019*). 5000 highly variable features (HVG) were used for normalization while accounting for percent mitochondrial reads. Variable features were defined and ranked by computing the variance of standardized gene counts after loess-based adjustment of mean–variance relationships (*Stuart et al., 2019*). Residuals of the fitted regression models were used as normalized gene expression values for HVGs. All libraries normalized via SCTransform were integrated using the canonical correlation analysis (CCA) pipeline in Seurat v3 to correct for batch effects and facilitate identification of similar cell identities across conditions. Highly ranked HVGs shared across all libraries were used as integration features. Integration anchors across libraries (correspondences of the selected features between cells) were computed over the first 30 CCA dimensions in the combined dataset and then used to inform the subsequent integration and grouping of cells. After integration, Seurat v3 was used for principal component analysis (PCA) and cell clustering. 100 principal components (PCs) of the integrated dataset were used for graph-based clustering and Louvain algorithm optimization as implemented in FindNeighbors and FindClusters. The final resolution in FindClusters was set to resolution = 2, yielding 96 cell clusters in our dataset. We selected this resolution to replicate the clustering pattern of a similarly processed *Drosophila* whole-brain scRNA-seq dataset (*Davie et al., 2018*). 100 PCs were used to embed cells in 2D space via uniform manifold approximation and projection (UMAP). Normalization of gene counts used in differential expression analysis, cell cluster marker gene computation, cell identity annotation, and other applications directly comparing gene expression levels between cell clusters were computed separately on the non-integrated gene expression data using the NormalizeData function in Seurat v3. In brief, for each gene in each cell, unique molecular identifiers (UMI) were divided by the sum UMIs in that cell, multiplied by a scalar (10,000), and log transformed. However, cell cluster membership (clusters 0–95) was defined using the integrated dataset as described above. The six additional libraries that comprise the day 10 replication dataset were clustered, integrated, and analyzed separately using the identical pipeline.

## Doublet detection

DoubletFinder was applied per library to predict and remove heterotypic doublets, leaving a total of 48,111 high-quality single cells in the discovery dataset. For each library, artificial doublets were generated from the existing data. PCA was performed after merging the real and artificial data and a distance matrix was generated with the first 40 PCs to compute the proportion of artificial K-nearest-neighbors (pANN) for each cell. PC neighborhood size (pK) for computing pANN was estimated for each library as previously described (*McGinnis et al., 2019*). The number of suspected doublets per library was estimated and cells were ranked by pANN for removal. Total doublet proportion for each library was computed based on a custom linear equation of the input-to-multiplet estimation provided by the 10x Chromium documentation: $Y = 5.272 \times 10^{-4} + 7.589 \times 10^{-6} (x)$, x being the number of recovered intact cells after the initial filtering criteria described above. The linear equation was generated based on recovery estimations in the manufacturer's protocol. Adjustment of the estimated doublet proportion for undetectable homotypic doublets was applied in DoubletFinder by using the Seurat clustering classifications at resolution = 2 as described above.

## SCENIC regulons

Gene regulatory networks (regulons) were computed using the Python implementation of SCENIC (pySCENIC). Raw gene abundances (UMIs) for 48,111 high-quality cells were exported as a loom object via loompy, and pySCENIC was implemented as described in *Van de Sande et al., 2020*. Putative gene targets for the published list of 815 *Drosophila* transcription factors (TFs) (see Key Resources Table) were inferred by tree-based regression (GRNBoost2) where expression of each gene was regressed on TFs, producing a list of adjacencies connecting TFs to their target genes (non-mutually exclusive). In the cisTarget step, modules were retained for further analysis if the regulatory motif of its parent TF was enriched among most gene members. Within retained modules, genes lacking enrichment of the appropriate motif were pruned. TF-motif annotations and pre-computed motif-gene rankings were obtained from https://resources.aertslab.org/cistarget/, *Drosophila* v8; motif search space encompassed up to 5 kb upstream of transcription start sites and intronic regions. This pipeline identified 183 regulons, encompassing 7134 out of 14,907 genes in the transcriptome dataset (*Figure 4—source data 2*), and cell-level activity for each regulon was computed by a ranking and recovery approach using pySCENIC AUCell. Within each cell, genes were ranked by expression level in a descending order, then the cumulative number of genes recovered belonging to a regulon at each rank was recorded. An area under the curve (AUC) was calculated after applying a default cutoff at the 95th percentile of gene ranks and is used to infer regulon activity. High AUC scores indicate greater representation of a given regulon among the top 5% of highly expressed genes in a cell. AUC scores for the 183 regulons across 48,111 cells were used for unsupervised clustering by UMAP for visualization of cell relationships based on gene regulatory networks (*Figure 4—figure supplement 4A*).

## Annotation of cell identity/abundance

We searched for cell identities of the 96 defined clusters by consolidating a series of four analytic approaches (*Figure 1—figure supplement 2A*). Two published datasets were used as references for our annotation procedure, including 56,902 cells from adult wildtype *Drosophila* (*w*$^{1118}$ and *DGRP-551*) brains profiled at days 0, 1, 3, 6, 9, 15, 30, and 50 (*Davie et al., 2018*), as well as 109,743 cells from adult Canton-S *Drosophila* optic lobes at day 3 (*Özel et al., 2021*). Cell clusters in these references were previously annotated using available literature-based cell markers or statistical inference with published bulk RNA-sequencing of reporter-targeted cell types. The Davie et al. dataset contained 87 cell clusters (Seurat FindClusters res = 2.0) with 41 assigned cell identities. The *Özel et al., 2021* dataset was clustered at a higher resolution (Seurat FindClusters res = 10), containing 200 cell clusters and 87 assigned cell identities. First, Scmap-cluster was used to compute gene expression correlation between each cell in our dataset to all defined clusters in the Davie and Özel datasets. 500 genes with higher-than-expected dropouts were selected as correlation features as described in *Kiselev et al., 2018*. The cosine similarity, Spearman and Pearson correlations of these features were subsequently computed between each cell in our dataset and all reference cluster centroids. For a cell to be mapped to a reference cluster, two out of three similarity scores must be concordant, and at least one must be greater than 0.7. Second, we intersected the top 20 cluster markers for each cell cluster (ranked by log2 fold change) in our dataset with the top 20 markers in each reference cluster. Cluster markers (cluster-enriched genes) for our 96 cell clusters were computed by differential expression analysis of the non-integrated, normalized gene abundances, comparing each cell cluster against all remaining cells. Markers were defined as positively differentially expressed genes (log2 fold-change greater than 0.1, Benjamini–Hochberg [BH]-corrected p-value 0.05) when comparing cells in given cluster versus all remaining cells in the dataset. Cell clusters were 'mapped' to a reference cluster in the Davie dataset (whole brain reference) if at least 13/20 top markers were shared. Likewise, a cluster was mapped in the Özel dataset (optic lobe reference) if at least 7/20 top markers were shared. These cutoffs were empirically determined by maximizing the number of best matches. Cell cluster markers for our dataset are listed in *Figure 1—source data 2*. Third, a trained neural network classifier for adult neurons as described in *Özel et al., 2021* was implemented in Python to label optic lobe neurons in our dataset. Log-normalized expression of 533 genes (out of the 587 genes in the Özel adult training set) across all cells were used as the input for the classifier. Finally, we checked for positive expression of well-established cell markers in each cluster (*Figure 1—source data 4*, *Figure 1—figure supplement 2B*, and *Figure 1—figure supplement 3*), using published cell marker datasets

(*Davie et al., 2018*; *Konstantinides et al., 2018*; *Kremer et al., 2017*). Most cell cluster annotations were evaluated and consolidated based on best agreement across two or more approaches within or across the Davie and Özel references. Less certain annotations were visually inspected in UMAP space to check for proximity with adjacent clusters and manually evaluated for cell marker expression. Cell-level confidence for scmap assignment (similarity score) or neural network classifications (confidence score) were also manually evaluated. Results from the Özel reference was prioritized for optic lobe neurons, especially for cell clusters that may be heterogeneous in the Davie reference (Dm8/Tm5c, TmY14, Tm9, Tm5ab, Mt1). Pm neurons, chiasm glia, and subperineurial glia did not reach consensus across two or more approaches and were thus deemed less confident annotations. Several other optic lobe cell types were well mapped in a single approach to the *Özel et al., 2021* dataset (TmY8, TmY3, Tm5c, Tm5ab, Tm20, Dm2, Dm8, Mi9, LC12, and LC17), where robust metrics were observed from the optic lobe neural network predictor or with scmap. Confirming our cell identity correspondences with the published scRNAseq datasets, we found high correlation among normalized gene expression when comparing individual cells in our dataset with the cluster-level means of the transcriptome in reference clusters as computed by cosine similarity (*Figure 1—figure supplements 1 and 2C and D*). The cosine similarity score between each annotated cell in our dataset and its cluster-level counterpart in the Davie or Özel references were computed based on shared non-dropout (count >0) genes, that is, the transcriptome of each cell in our data was correlated to the cluster-level mean of corresponding genes in a reference cluster. Lower similarity scores may reflect gene expression changes induced by tau pathology, less confident annotation (in this study or in the references used), clustering resolution differences, or high variance in the reference cluster.

To annotate the replication scRNAseq data (69,128 cells), labels from the completed dataset above (48,111 cells) were transferred using the Seurat v3 FindTransferAnchors and TransferData functions. In brief, pairs of similar cells between the reference and query dataset were identified using a mutual nearest-neighbor approach after projecting the replication dataset onto the reference dataset in PCA-reduced space. The 5000 most variable genes in the new dataset were used for the dimensional reduction. Each cell was assigned a score and a predicted label from the reference dataset. Cell-level metadata for both the discovery and replication datasets are uploaded to the Synapse AMP-AD Knowledge Portal as noted in the Key Resources Table.

After annotation, cell counts for each assigned cluster (90 clusters) were first quantified per library (six libraries, ages: days 1, 10, and 20; genotypes: control, tau), and treated as count data. To adjust for extreme proportional differences in cell composition across libraries and differences in the total number of cells captured per library, cell counts were normalized using negative binomial generalized linear models (NB-GLM) as implemented in DESeq2 (*Love et al., 2014*). In brief, raw cell counts were modeled using NB-GLM with a fitted mean and a cluster-specific dispersion factor. Dispersion factors were computed based on mean count values using an empirical Bayes approach as described in *Love et al., 2014*. The fitted mean is composed of a library-specific size factor and a parameter proportional to the true counts in each cluster per library. To compute size factors per library, raw counts were organized in a matrix such that rows represent clusters and columns represent samples (libraries). Raw counts were first divided by the row-wise geometric means and then divided by the per-column median of resulting quotients (size factor) to obtain normalized cell count values per cluster. These normalized cell counts were used to generate the plots in *Figure 2B*. The three age groups for each genotype (days 1, 10, 20) were combined to produce an n = 3 comparison of cell counts between tau and control animals.

## Deconvolution of fly RNA-sequencing data

Deconvolving bulk-tissue RNA-sequencing data into estimated proportions of cell populations was performed by implementing Multi-subject Single-cell Deconvolution (*Wang et al., 2019*) using default parameters. MuSiC leverages cell-specific expression data from annotated scRNA-seq datasets and weighted non-negative least-squares regression to characterize cell compositions of bulk tissue gene expression data. This approach accounted for gene expression variability across samples and cells, thus upweighting the most consistently expressed genes across samples or cells for deconvolution. Whole-head RNA-sequencing counts of experimental conditions identical to those in this study were taken from *Mangleburg et al., 2020* and used as input for deconvolution. Specifically, cell counts for n = 2 control and n = 3 tau$^{R406W}$ samples at days 1, 10, and 20 were deconvolved (15 samples total,

each sample is a homogenate of 100 heads). 56,902 cells from a published *Drosophila* whole-brain scRNAseq dataset was used as an orthogonal reference for deconvolution, providing cell-specific transcriptional profiles from wildtype control animals ($w^{1118}$ and *DGRP-551*) (*Davie et al., 2018*). Individual scRNA libraries were treated as subjects in the MuSiC pipeline for evaluating gene expression variability in marker gene weighting. Both annotated and unannotated cell clusters in the reference scRNAseq dataset were included in the deconvolution pipeline. Select cell clusters with non-zero estimated proportions across two or more timepoints are plotted in *Figure 2—figure supplement 2*.

## Immunofluorescence and confocal microscopy

10-day-old female controls (*elav-GAL4/+*) or *elav>tau^R406W* (*elav-GAL4/+; UAS-tau^R406W/+*) were used for glial quantification immunofluorescence experiments. The animals were anesthetized with $CO_2$ and brains were dissected with forceps and fixed in 4% paraformaldehyde (PFA) overnight at 4°C. After fixation, PFA was aspirated and replaced with PBS with 2% Triton-X (PBST) and incubated at 4°C overnight for tissue penetration. Residual air trapped in brain tissues were removed by placing samples under a vacuum for 1 hr at room temperature. The brains were then incubated in blocking solution (5% normal goat serum in PBST) at room temperature, rocking for 1 hr. Primary antibodies were diluted in 0.3% PBST and samples were incubated in primary at 4°C, rocking for at least 24 hr. The primary antibody solution was aspirated, and the samples were washed with PBST (two quick washes followed by three 15 min washes). Samples were incubated in secondary antibodies at room temperature, rocking for 2 hr. The secondary antibody solution was then aspirated and the samples were washed with PBST (two quick washes followed by three 15 min washes). DAPI stain, when applicable, was added in the secondary antibody step. Whole brains were then mounted in Vectashield antifade mounting medium (Vector Laboratories, H-1000-10) and stored in the dark at 4°C until imaged. Samples were imaged on a Leica Microsystems SP8X confocal microscope. Z-stacks covered the entirety of whole-mount brains. We used the following antibodies and dilutions: mouse anti-Repo (8D12, 1:500 for glial quantification experiment, 1:50 for Rel experiment, DSHB), rat anti-Elav (7E8A10, 1:100, DSHB); rabbit anti-GFP (1:500; GeneTex), mouse anti-Rel (21F3, 1:500, DSHB), conjugated A488-Phalloidin (1:500; Thermo Fisher), Cy3 AffiniPure goat anti-mouse (H+L) (1:500; Jackson ImmunoResearch Laboratories), Alexa 647-conjugated goat anti-Rabbit IgG (1:500; Jackson ImmunoResearch), Alexa Fluor 488 donkey anti-mouse IgG (H+L) (1:500; Jackson ImmunoResearch), and Cy3 AffiniPure goat anti-rat IgG (H+L) (1:500; Jackson ImmunoResearch).

Quantification of glia from confocal immunofluorescence digital microscopy was performed using Imaris (v9.9.1) imaging software. We counted the total number of repo-positive cells using the 'spots' object and automatic detection parameters with local thresholding and background subtraction. Brain volume was determined by using the 'surfaces' object on the Phalloidin channel to encompass the entire three-dimensional volume of the brain. Graphs of raw repo-positive counts per brain as well as glial density (repo-positive counts divided by brain volume) were created in GraphPad Prism (v9.4.1) software. Glial quantifications were performed using full Z-stacks of whole-mount brains. Welch's *t*-test was used for comparisons between control and *tau^R406W* animals (n = 9 animals per group). The significance threshold was set to $p<0.05$. Error bars represent the 95% confidence interval.

Mean pixel intensity for DAPI, phalloidin, or repo signal was calculated for n = 9 brains per genotype using ImageJ/Fiji (units: corrected total cell fluorescence [CTCF]). In GraphPad Prism, the mean pixel intensity for each channel (DAPI, phalloidin, or repo) was compared between control vs. tau-expressing animals using parametric, unpaired, two-tailed *t*-tests. All experimental groups passed the Shapiro–Wilk test for normality except for phalloidin intensity in tau-expressing animals, so this comparison (mean phalloidin intensity in control vs. tau-expressing animals) was done with a nonparametric (Mann–Whitney) *t*-test. The BIOP JACoP plugin on ImageJ/Fiji was used to calculate colocalization between relish-GFP signal and elav or repo signal. Area of overlap between signals (in pixels) was calculated for each slice (n = 85 slices total) using a stack histogram as the threshold. The area of overlap was then divided by the total elav or repo area to find the percentage of elav or repo area that was also positive for relish-GFP signal.

## Bulk-tissue RNA-sequencing data

Bulk-RNA sequencing data and WGCNA co-expression modules of the experimental conditions described in this study were obtained from *Mangleburg et al., 2020*. WGCNA module expression

activity in scRNAseq was computed by taking the mean of module member genes within each cell. Cluster-level expression activity of WGCNA modules was then estimated by averaging the cell-level activity across all cells in a given cluster and was subsequently used to compute the log2 fold change of tau versus control expression activity.

### *Drosophila* NFκB signaling mediators

A list of *Drosophila* NFκB signaling pathway members was generated based on manual curation from published studies (*Valanne et al., 2011*; *Myllymäki et al., 2014*; *Kounatidis et al., 2017*; *Li et al., 2020*) and included the following genes: *PGRP-LE, imd, Tak1, key, Rel, eff, PGRP-LC, bsk, akirin, Jra, sick, Tab2, IKKbeta, Uev1A, ben, Dredd, Fadd, PGRP-LA, Diap2, Diap1, egr, Traf6, trbd, pirk, casp, PGRP-LB, PGRP-LF, dnr1, scny, RYBP, PGRP-SC1a, PGRP-SC1b, PGRP-SC2, CYLD, POSH, spirit, spheroide, spz, PGRP-SA, PGRP-SD, pll, Myd88, dl, Gprk2, Deaf1, Tl, psh, grass, modSP, Dif, mop, tub, cact, nec, Pli, 18w, Toll-4, Tehao, Toll-6, Toll-7, Tollo*, and *Toll-9*.

### Analysis of differential cellular abundance

Statistical testing of the $\log_2$ fold change (log2FC) of tau versus control normalized cell abundance was performed using negative binomial-generalized linear models (NB-GLM) as implemented in DESeq2. Age was treated as a covariate, and a Wald test was performed on the coefficient of the genotype variable using the following model: $\log_2$(cell count) ~ age + genotype. Using DESeq2, log2FC was computed for each cluster (*elav>tau^{R406W}* vs. control) based on maximum-likelihood estimation after fitting the GLM. Raw log2FC values were transformed using an adaptive shrinkage estimator from the 'ashr' R package as implemented in DESeq2 to account for clusters with high dispersion or low counts. These transformed log2FC values were then used for cell abundance analysis and interpretation. A BH-adjusted p-value<0.05 was used to establish significance of Wald test statistic. In order to generate the plots for *Figure 2B*, normalized cell counts were obtained using the *counts* function in DESeq2. Results were visualized using box and whisker plots, including the following values: median, minimum/maximum, and lower/upper quartiles.

In order to better understand how relative changes might influence cell abundance estimates, we inferred confidence intervals for cell cluster log2FC values (*Figure 2—figure supplement 3A*). Based on experimental ground truth (*Serrano-Pozo et al., 2013*), the log2FC value for seven cell clusters (Ensheathing glia, Perineurial glia, Astrocyte-like glia, Cortex glia, Chiasm glia, Subperineurial glia, and Hemocytes) was centered to zero. Specifically, the value of each glial cluster was iteratively subtracted from the log2FC values for all other clusters, establishing a minimum and maximum log2FC value for all cell clusters. We predict that the true cell abundance falls within this computed range, after accounting for potential proportional influences. A range that includes zero thus suggests there may be no true change between tau and control.

### Analysis of differential gene expression

Differential gene expression analyses were performed using Model-based Analysis of Single-cell Transcriptomics (MAST) for each cell cluster (*Finak et al., 2015*). In brief, generalized linear hurdle models were used to compute differential expression, where logistic regression was used to account for stochastic dropouts, and a Gaussian linear model was fitted to predict gene expression levels. Differential expression was determined by a likelihood ratio test. We required that differentially expressed genes meet a significance threshold of BH-adjusted p-value<0.05; absolute log2 fold-change > 0.1; and detectable (non-zero) expression in at least 10% of cells in the cluster. Cellular detection rate (CDR, fraction of genes reliably detected in each cell) was included as a covariate in all regression models, as in published protocols (*Finak et al., 2015*). CDR acts as a proxy for estimating the effect of dropout events, amplification efficiency, cell volume, and other extrinsic factors while performing expression-related regression analyses. Analyses of tau-induced differential expression (age-adjusted) also included age as a regression model covariate. Separately, aging-induced changes within each cell cluster were computed from control data (*elav-GAL4/+*), comparing differential gene expression between days 1 and 10, and days 10 and 20. In order to evaluate robustness and replicability, cross-sectional, tau-induced differentially expressed genes were also computed in day 10 animals (tau vs. control) in the replication data and results were compared between the discovery and replication datasets. Similarly, cross-sectional tau-induced differential expression was also computed

for each timepoint in the discovery dataset. For differential expression of the human cell subclusters reported in *Mathys et al., 2019*, normalized counts between individuals with AD pathology (n = 24) and low/no pathology (n = 24) were compared using MAST for each cell subcluster as described above.

To assess cell-type-specific differences in regulon gene expression levels (*Figure 4C*), we performed differential regulon analysis using base statistical packages in R. In brief, mean expression of gene members in each regulon were computed per cell, and cell-type-matched comparisons were made between control and *elav>tau$^{R406W}$* using linear regression, including age as a covariate. Specifically, for each cell type, a likelihood ratio test compared the fit of a full model (Regulon Expression ~ Genotype + age) and a reduced model (Regulon Expression ~ age), evaluating the contribution of genotype to model fit. Significance was set at a BH-adjusted p-value<0.05. Regulon log$_2$-fold changes were computed for *elav>tau$^{R406W}$* versus control mean expression in each cluster.

Overrepresentation analysis (ORA) of differentially expressed gene sets were completed using the R implementation of WEBGESTALT (*Wang et al., 2013*). The following databases were used: Gene Ontology (GO) biological processes, GO molecular functions, GO cellular component, KEGG, and Panther. Enrichment significance was defined by hypergeometric test, followed by p-value adjustment using the BH-procedure; significance was set at p<0.05. ORA of the tau unique gene set (n = 363 genes) was performed using gProfiler (*Raudvere et al., 2019*). The Gene Ontology (GO), Human phenotype ontology (HP), KEGG, miRTarBase (MIRNA), Transfac (TF), and WikiPathways (WP) databases were used for querying genes. The organism parameter was set to 'dmelanogaster' and the 'fdr' correction method was used to apply the BH multiple testing correction. A false discovery rate (FDR) < 0.05 was the threshold for significance.

## Multiple regression with elastic net

To identify features driving cell vulnerability in our scRNAseq dataset, we pooled information across cell clusters by performing elastic net regression. For all clusters showing significant *elav>tau$^{R406W}$* vs. control cell abundance changes, log2FC values were regressed on the cell-type-specific mean expression for 183 regulons. Given our goal to identify the factors that influence cell-type-specific vulnerability, we focused on eight cell clusters with significant cell loss (FDR < 0.05). In a secondary analysis, we repeated elastic net regression and considered a larger number of potential predictor variables including (i) the 183 regulons (as above); (ii) 2793 unique GO, KEGG, and Panther pathways found to be significantly enriched among *elav>tau$^{R406W}$* differentially expressed genes; (iii) 7 WGCNA modules altered in *elav>tau$^{R406W}$* (*Mangleburg et al., 2020*); and (iv) curated NFκB signaling pathways. In addition, we also considered a large number of (v) cell cluster technical parameters as potential predictors, including normalized cell counts, mean tau transgene expression, sum of UMIs, mean percent mitochondrial reads, and number of tau-induced differentially expressed genes (age-adjusted). For this analysis, all computed variables (e.g., GO pathways, WGCNA modules, regulons, NFκB genes) were first averaged within each cell, then averaged across all cells in order to determine a mean value for each cell cluster. Cluster-level means for all gene sets were computed using pooled cell data from both *elav>tau$^{R406W}$* and controls and all ages. For gene sets derived from ORA analyses, we restricted consideration to those differentially expressed genes driving enrichment. We generated a matrix consisting of rows for each cell cluster and columns with values/means for each potential predictor variable.

We used the *caret* and *glmnet* packages in R to organize the data and perform elastic net regularized regression. Alpha (ridge vs. lasso characteristic) and lambda (shrinkage parameter) values were tuned in a 1000 × 1000 grid using repeated threefold cross-validation in caret, and the average root mean squared errors (RMSE) from testing the partitions were used to assess model performance. Threefold cross-validation was repeated 100 times for all alpha-lambda pairs using a different data fold split for each iteration in order to account for variability in model performance from random sample partitioning. The mean of all prediction errors was used to assess the final performance of each alpha-lambda pair, and we selected the model with the lowest RMSE. Lastly, we generated a ranklist of predictor variables for tau-induced cell abundance changes based on the magnitude of coefficients from the selected model (*Figure 4—figure supplement 7B*, *Figure 4—source data 4*).

## Cross-species analysis with human and mouse RNAseq datasets

Previously published human snRNAseq data (*Mathys et al., 2019*; *Grubman et al., 2019*) were reprocessed and filtered using the identical pipeline as describe above for *Drosophila* data. Raw gene counts were normalized using the NormalizeData function in Seurat v3. The resulting 70,634 filtered cells from the Mathys data were re-clustered using our pipeline above for visual representation in *Figure 5*; however, the cell cluster annotations from the original publication were preserved. 13,214 filtered cells from the Grubman data were similarly promoted for analysis. *Drosophila* orthologs of genes detected in human datasets were determined using the DRSC Integrated Ortholog Prediction Tool (DIOPT) (*Hu et al., 2011*), requiring a minimum DIOPT score threshold of 5 or greater. If more than one fly ortholog was identified, we selected the ortholog with either (i) the highest DIOPT score, (ii) the highest weighted DIOPT score, or (iii) the highest ranked option (best score when mapped both forward and reverse). Thus, 5630 or 4145 human-fly gene ortholog pairs, respectively, were considered for cross-species analyses of the Mathys and Grubman datasets. We scaled normalized expression of each gene with mean = 0 and variance = 1. Cluster-level gene expression was computed by averaging scaled expression values from all cells. Subsequently, we performed Pearson correlation analysis for all cluster pairs to quantify transcriptional similarities between fly and human cell, examining pairwise correlation coefficients for all gene-orthologs across all clusters. For visualization, we generated heatmaps representing Pearson correlation coefficients by seriation with hierarchical clustering. Association statistics for human neuropathological traits (heatmap at top of *Figure 5A*) were repurposed directly from the published supplementary from *Mathys et al., 2019*. For quantification of overlap between human microglia and fly ensheathing glia, we examined conserved differentially expressed genes using the hypergeometric overlap test. To demonstrate control-only correlations, scRNAseq from the filtered Davie et al. dataset (subsetted for annotated cell types) were compared to snRNAseq profiles of 24 control individuals in the Mathys et al. data. To compute Rel regulon differential expression in a tauopathy mouse model, we used normalized scRNAseq pseudobulk counts from *MAPT* P301L mice (n = 3) and non-transgenic controls (n = 2) as published in *Lee et al., 2021*. Conserved mice genes (DIOPT > 4) in the fly Rel regulon were averaged per cell cluster (554 mice genes mapped to at least one fly ortholog) and a likelihood ratio test was used to evaluate the contribution of genotype to differential expression in each cluster.

## Acknowledgements

We are grateful to Pinghan Zhao, Tom Lee, Alma Perez, Katy Zhu, Akash Tarkunde and Bismark Amoh for assistance with *Drosophila* brain dissections. We also thank the Bloomington Drosophila Stock Center, the Vienna Drosophila RNAi Center, FlyBase (*Gramates et al., 2017*), the Developmental Studies Hybridoma Bank, and BioRender. This study was supported by grants from the National Institutes of Health (NIH) (R01AG057339, R01AG053960, U01AG061357, and U01AG046161). In addition, we utilized BCM core resources, including the Intellectual and Developmental Disabilities Research Center (P50HD103555), Genomic and RNA Profiling (S10OD023469), and Single Cell Genomics (S10OD025240 and Cancer Prevention Research Institute of Texas grant RP200504). HY is additionally supported by the Parkinson's Foundation (PF-PRF-830012) and the Alzheimer's Association (AARF-21-848017). JMS was additionally supported by the Huffington Foundation, McGee Family Foundation, the Jan and Dan Duncan Neurological Research Institute at Texas Children's Hospital, and The Effie Marie Caine Endowed Chair for Alzheimer's Research.

## Additional information

### Funding

| Funder | Grant reference number | Author |
| --- | --- | --- |
| National Institute on Aging | R01AG057339 | Zhandong Liu<br>Juan Botas<br>Joshua M Shulman |
| National Institute on Aging | R01AG053960 | Joshua M Shulman |

| Funder | Grant reference number | Author |
| --- | --- | --- |
| National Institute on Aging | U01AG061357 | Joshua M Shulman |
| National Institute on Aging | U01AG046161 | Joshua M Shulman |
| Eunice Kennedy Shriver National Institute of Child Health and Human Development | P50HD103555 | Joshua M Shulman |
| National Institutes of Health | S10OD023469 | Brandon T Pekarek |
| National Institutes of Health | S10OD025240 | Brandon T Pekarek |
| Cancer Prevention and Research Institute of Texas | RP200504 | Joshua M Shulman |
| Parkinson's Foundation | PF-PRF-830012 | Hui Ye |
| Huffington Foundation | | Zhandong Liu Juan Botas Joshua M Shulman |
| McGee Family Foundation | | Joshua M Shulman |
| Duncan Neurological Research Institute | | Zhandong Liu Ismael Al-Ramahi Juan Botas Joshua M Shulman |
| Effie Marie Caine Endowed Chair for Alzheimer's Research | | Joshua M Shulman |
| Alzheimer's Association | AARF-21-848017 | Hui Ye |

The funders had no role in study design, data collection and interpretation, or the decision to submit the work for publication.

## Author contributions

Timothy Wu, Conceptualization, Formal analysis, Investigation, Writing - original draft, Writing – review and editing; Jennifer M Deger, Hui Ye, Caiwei Guo, Investigation, Writing – review and editing; Justin Dhindsa, Brandon T Pekarek, Rami Al-Ouran, Formal analysis, Investigation, Writing – review and editing; Zhandong Liu, Juan Botas, Conceptualization, Supervision, Funding acquisition, Writing – review and editing; Ismael Al-Ramahi, Conceptualization, Supervision, Writing – review and editing; Joshua M Shulman, Conceptualization, Supervision, Funding acquisition, Writing - original draft, Writing – review and editing

## Author ORCIDs

Timothy Wu http://orcid.org/0000-0001-5296-2023
Hui Ye http://orcid.org/0000-0003-3965-9702
Juan Botas http://orcid.org/0000-0001-5476-5955
Joshua M Shulman http://orcid.org/0000-0002-1835-1971

## Decision letter and Author response

Decision letter https://doi.org/10.7554/eLife.85251.sa1
Author response https://doi.org/10.7554/eLife.85251.sa2

# Additional files

## Supplementary files
• MDAR checklist

## Data availability

All original single cell sequencing data have been uploaded to the Accelerating Medicines Parternship (AMP)-AD Knowledge Portal on Synapse and can be accessed through the DOI: https://doi.org/10.7303/syn35798807.1.

The following dataset was generated:

| Author(s) | Year | Dataset title | Dataset URL | Database and Identifier |
|---|---|---|---|---|
| Wu T, Deger JM, Ye H, Guo C, Dhindsa J, Pekarek BT, Al-Ouran R, Liu Z, Al-Ramahi I, Botas J, Shulman JM | 2022 | Tau polarizes an aging transcriptional signature to excitatory neurons and glia | https://doi.org/10.7303/syn35798807.1 | Synapse, 10.7303/syn35798807.1 |

The following previously published datasets were used:

| Author(s) | Year | Dataset title | Dataset URL | Database and Identifier |
|---|---|---|---|---|
| Mathys H | 2019 | The Single-cell transcriptomic analysis of Alzheimer's disease (snRNAseqPFC_BA10) Study | https://doi.org/10.7303/syn18485175 | Synapse, 10.7303/syn18485175 |
| Grubman, et al | 2019 | A single-cell atlas of the human cortex reveals drivers of transcriptional changes in Alzheimer's disease in specific cell subpopulations | https://www.ncbi.nlm.nih.gov/geo/query/acc.cgi?acc=GSE138852 | NCBI Gene Expression Omnibus, GSE138852 |
| Davie, et al | 2018 | A single-cell transcriptome atlas of the ageing *Drosophila* brain | https://www.ncbi.nlm.nih.gov/geo/query/acc.cgi?acc=GSE107451 | NCBI Gene Expression Omnibus, GSE107451 |
| Ozel, et al | 2020 | Neuronal diversity and convergence in a visual system developmental atlas | https://www.ncbi.nlm.nih.gov/geo/query/acc.cgi?acc=GSE142789 | NCBI Gene Expression Omnibus, GSE142789 |
| Mangleburg, et al | 2020 | Integrated analysis of the aging brain transcriptome and proteome in tauopathy | https://doi.org/10.7303/syn7274101 | Synapse, 10.7303/syn7274101 |
| Van de Sande B | 2020 | pySCENIC: List of *Drosophila* transcription factors | https://github.com/aertslab/pySCENIC/blob/master/resources/allTFs_dmel.txt | Aerts lab resource, allTFs_dmel |
| Van de Sande B | 2020 | pySCENIC: Motif-Gene rankings | https://resources.aertslab.org/cistarget/databases/old/drosophila_melanogaster/dm6/flybase_r6.02/mc8nr/gene_based/dm6-5kb-upstream-full-tx-11species.mc8nr.feather | Aerts lab resource, mc8nr.feather |
| Van de Sande B | 2020 | psySCEnIC: Transcription factor to motif annotations | https://resources.aertslab.org/cistarget/motif2tf/motifs-v8-nr.flybase-m0.001-o0.0.tbl | Aerts lab resource, m0.001-o0.0.tbl |

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
