## [Editor Report]

Wu et al. have provided a revised manuscript that presents important new findings that start to explain cell type vulnerability and the types of transcriptional changes that occur in the context of neurodegenerative diseases. They cleverly use *Drosophila* for this as they have access to numerous brain cells and exquisite genetic control. They present compelling evidence of transcriptional deregulation and affected pathways in relation to Tau toxicity in a well-controlled study. They also tested if affected pathways modify toxicity but were not successful, however, as pointed out, this can have different reasons. This paper is of broad interest to those in the field of neurodegeneration and neuronal disease and from a methodological point of view to single-cell biologists.

---

## [Decision Letter]

**Decision letter after peer review:**

Thank you for submitting your article "Tau polarizes an aging transcriptional signature to excitatory neurons and glia" for consideration by *eLife* and please accept our apologies for the longer than usual reviewing time. Your article has been reviewed by 3 peer reviewers, one of whom is a member of our Board of Reviewing Editors, and the evaluation has been overseen by Claude Desplan as the Senior Editor. The following individual involved in review of your submission has agreed to reveal their identity: Simon G Sprecher (Reviewer #3).

Essential revisions:

1) Given that all vulnerable cell types were already lost at day 1, the reviewers were unclear whether the model assesses age-dependent neurodegeneration. This may also be developmental toxicity. There should be a balanced discussion on this or alternatively, data could be included making use of models that show defects only at older age.

2) There were some concerns about genetic background (cf rev 1) and controls (cf rev 2): ie is there a possibility to include wt-tau or carefully discussing this; likewise, given the depth of analysis one achieves with single cell seq approaches, genetic background issues can be real confounding factors. Was this addressed in the experimental design.

3) The finding of involvement of the NFkB pathway is interesting, but causality has not been shown. All reviewers thought it would be rather simple to put the idea to test by genetically modulating this pathway and assessing if neuronal loss is rescued.

4) the last comment by reviewer 2 was also deemed important. The comparison between species and of an FTD-Tau mutation with AD needs to be toned down.

5) the other issues can likely be addressed by textual changes or added discussion.

*Reviewer #1 (Recommendations for the authors):*

I have these points that would improve the paper:

– Can the authors test whether the neuronal loss in their model is due neurodegeneration rather than developmental toxicity to tau.

– Given that they find the NFkB pathway to be involved in tauopathy in a model organism, it would be fascinating if they put this idea to test and show causality by genetically modulating this pathway and rescuing neuronal loss.

– Can the authors please mention the genetic background of all lines used in the Methods. Where UAS-tau and the wild-type fly that was crossed to elav-Gal4 to serve as a control in the same genetic background?

– The authors should include in the counting step whatever 3' UTR the tau transformation vector had that was used for generating the fly model, since the fast majority of reads should map there rather than in the tau sequence. For now, it seems that NM_016834.5:151-1302 (Methods) represents the CDS.

– Figure S6B and S10D are missing a quantification. In addition, for 10D it would be helpful to add a negative control to see how specific the signal is, such as a control fly without the endogenous GFP tag.

– It is interesting that glia seem to react strongly to tau. However, it is not clear whether this is cell autonomous – because they also express tau – or as a reaction of the neuronal tau expression. Is the promoter they use neuronal and would we expect any expression in glia? Can they maybe add a panel to Figure S8 with a boxplot showing tau expression levels in glia cell types and neurons.

– It is interesting how the authors find multiple regulons (some with >2x larger coefficients than the Rel regulon) to be associated with the degree of vulnerability. For the curious reader it would be helpful to at least point them out and briefly mention the underlying biology.

– Why were in Figure S7 KEGG pathways only annotated for few cell types. This should be explained in legend or annotated more widely, e.g. in all cell types that are lost.

– In Figure S9B not all cell-types shown. Does this mean that no pathways were found in those, maybe they could add this to the legend? And why are the KEGG terms in Figure S7B for a'/b'-KC different than in S9B?

Also a typo is in this legend: 'including pathways that are actively in cell-type specific vs. more global patterns'

– Figure S8 should add whether this is counts or log-scale.

– Figure 4D can the authors add explicitly whether this is control and tau cells pooled?

*Reviewer #2 (Recommendations for the authors):*

Wu et al. conducted longitudinal single-nucleus RNA sequencing in a *Drosophila* transgenic line expressing pathogenic tau (Arg406 ->Trp) and control to study presenile degenerative dementia with bitemporal atrophy. Their data is consistent with previous findings on Tau neurotoxicity, which significantly affects excitatory neurons in human brain samples and transgenic mice. Intriguingly, intracellular transgenic Tau induced strong transcriptional signatures, aging-like signatures, and an innate immune response, including the NFKB pathway, in the transgenic animals. This dataset provides a valuable resource for exploring dynamic, age-dependent gene expression changes at a cellular level. The authors propose that innate immune signatures may serve as predictors of neuronal subtype vulnerability in tauopathies. However, the observed skewing of cell proportions in day-1 animals necessitates stronger evidence to support this hypothesis, which is currently lacking in the manuscript. The paper is primarily descriptive and lacks mechanistic insights. Furthermore, the identified pathways/genes presented in the paper lack orthogonal validation.

1. About the controls: Authors compared Tau transgenic line (Arg406 ->Trp) with the control (GAL4 expressing animals) but not with wt-tau line. They may potentially lead to misinterpretation. Although Wittmann et al. 2001 noted toxicity when wt-tau is expressed, the toxicity is much less compared to Tau transgenic line. Or would another alternative be to use mutant tau animals lacking aggregation-prone regions?

2. It is striking to see the drastic cell proportion changes on day-1 (figure 2B), which may reflect the deficits in neuronal development. Did authors check the expression of transgene expression levels across neuronal subtypes to make sure the vulnerability is not due to a difference in the Tau transgene expression?

3. Figure 2B-D, although authors admit that the difference is likely due to the "increase in glial cell abundance from scRNAseq is likely a consequence of proportional changes in single cell suspensions due to neuronal loss," is there a way to quantitatively assess this? Especially authors know the amount of neuronal loss and increase in the glial cells through scRNAseq.

4. Line 204-205, authors claim, "93% of tau-induced differentially expressed genes were also triggered by aging in control flies". However, figure 3A does not reflect the 93% similarity. There are more DEGs in age-specific conditions compared to the Tau. The same holds true for the Figure 2B. Moreover

5. Figure 2B, the number of DEG in cluster Lai and Kenyon cells is highly skewed in the Tau transgenic lines. I find it is intriguing to see high number of DEGs in the cells that are degenerating. Since these plots don't tell much about whether they are up or down, it would be good to mention what proportion of the genes are up and down.

6. The authors claim that among non-neuronal cell types, ensheathing glia, cortex glia, astrocyte-like glia, and hemocytes have the highest number of tau-driven DEGs, but this is not clear from the UMAP in Figure 3C. Additionally, Figure 3C lacks a scale bar, making it difficult to interpret and compare the figure with Figure 3B.

7. In line 249, the authors claim 90% concordance with previously published datasets, but the data representing this is missing in the paper. Additionally, performing DEG with pseudo-bulk from different clusters and performing DEG to find the concordance may not be very informative. For example, did the authors find consistent gene signatures per cluster when compared with previous datasets? This data should be provided.

8. Authors have created an excellent data resource, and it would be interesting to explore the resilience of inhibitory neurons or the vulnerability of excitatory neurons to gain more insights into the cell-type-specific vulnerability or resilience mechanisms. The authors should present a couple of volcano plots showing the differentially expressed genes between important clusters, such as LAI, Kenyon cells, ensheathing glia, etc.

9. It would be beneficial for the authors to explore these pathways in greater depth and perform further experimental validation to strengthen the findings using orthogonal approaches. For instance, a rescue experiment where NFKB/Relish is knocked out to see if this modifies Tau toxicity.

10. In Figure 5, the authors compared cell-type-specific transcriptional signatures between human Alzheimer's disease (AD) and *Drosophila*. However, some readers may find this comparison difficult to comprehend as the two species differ vastly. Moreover, the Tau mutation that the authors investigated is not associated with AD. Also, in AD, amyloid pathology significantly drives gene expression in immune cells, which is absent in *Drosophila*. Authors should consider taking the relevant dataset derived from the Arg406 ->Trp patients or from the iPSC-derived cells to validate the observations.

*Reviewer #3 (Recommendations for the authors):*

As mentioned above, I feel this is an elegant and nicely described study. It provides a further example of the power of single-cell transcriptomic to assess disease genes, showing that in such a fashion one can assess the impact of cell-types, but also the actual genes that are altered. It also shows that the findings can be transferred to patient tissue, thus integrating previous published human sc-data.

There are a few points that I feel might be important to take into account.

– The authors show that ratios of cells are changed in response to tau-GOF, however no explanation is given. What is the basis of this alteration? Cell-death, changes of differentiation/proliferation (it seems the GOF is throughout development in "elav" cells).

– The integration and comparison of fly/human data is a bit short, I could not follow the process of how this was achieved, what framework the authors used etc.

– Conceptually, I think it is nice to see the in-silico analysis of the tau-GOF and aging, however I feel some in vivo validation might have been beneficial to support the claims of affected cell-types and differential expressed genes.

While I fully agree that an extensive genetic analysis may be beyond the scope of the current paper, a proof-of-concept analysis would have been supportive of the validity of the data.

---

## [Author Response]

Essential revisions:1) Given that all vulnerable cell types were already lost at day 1, the reviewers were unclear whether the model assesses age-dependent neurodegeneration. This may also be developmental toxicity. There should be a balanced discussion on this or alternatively, data could be included making use of models that show defects only at older age.

The reviewers raise an important point, and we agree that “developmental toxicity” may contribute to some of our observations. The *elav-GAL4* pan-neuronal driver activates expression of *tau^R406W^* during developmental stages and observed changes in cell-abundance or gene expression may reflect this.

For example, *tau* developmental toxicity has been shown to cause malformation of mushroom body structures (Kosmidis et al., 2010), and this phenotype likely explains the reductions in several cell clusters in our dataset. However, the widely-used *elav>tau^R406W^* model also recapitulates aging-dependent, progressive neuronal loss and CNS dysfunction (Wittmann et al., 2001). Consequently, our longitudinal design can additionally highlight the accompanying cell-specific transcriptional changes. While we acknowledge that most of the changes in cell abundance highlighted in Figure 2B appear to be established at day 1, in many cases, age-dependent changes are also strongly suggested (e.g. Cluster 1, 9, and 12, along with astrocyte-like glia). One important caveat is that the dataset lacks replicate samples at either day 1 or day 20. Therefore, while changes in cell abundance at these timepoints are suggestive in many cases, our experimental design does not permit cross-sectional statistical analysis at either timepoint, nor can we quantitatively examine changes over time (e.g. 1 vs. 10 days or 1 vs. 20 days). For this reason, our analyses of cell abundance (Figure 2A) relied on pooled data from all 3 time points. We also performed a replication analysis at 10 day (Figure 2—figure supplement 1A), based an independent generated dataset with triplicate samples. In a complementary analysis, we also employed the deconvolution algorithm MuSiC to examine cell type proportions from *elav>tau^R406W^* bulk RNAseq data (Figure 2—figure supplement 2). Notably, our experimental design *does* permit robust analysis of age-dependent changes in cell-specific gene expression. Our primary analysis (Figure 3B) leverages the longitudinal data and includes adjustment for age as a covariate. However, in order to address the reviewer’s feedback, we have now included cross-sectional analyses of cell-specific differential expression changes (Results text and Figure 3–Source Data 5). These new data readily allow interrogation of specific genes and pathways for mapping of dynamic changes over time. For example, in our revisions, we note that tau-induced Relish regulon activity is amplified (or attenuated) in selected cell types with aging, highlighting potential dynamic changes in immune signaling pathways (Results text and Figure 4—figure supplement 5). For example, in the L1-5 lamina neuron cluster, NFkB responsive genes are only significantly differentially expressed at 20-days. Similarly, the innate-immune signature in astrocyte-like glia appears driven by changes in 10- and 20-day-old flies; no significant change is observed at 1-day. As requested, we have also added text to the results and discussio in order to provide a more balanced and nuanced discussion of these issues and emphasizing the challenge to disentangle developmental toxicity from age-dependent neurodegeneration. We also include new discussion of a recently published study that appeared while our manuscript was under review, which uses the *nsyb>tau^P301L^* model expected to have more restricted expression within the adult brain (Praschberger et al., *Neuron* 2023, PMID: 36948206). Finally, we have also added text to the Figure 2 legend clarifying that significant differences in tau-induced cell abundance were based on comparisons of pooled timepoints.

2) There were some concerns about genetic background (cf rev 1) and controls (cf rev 2): ie is there a possibility to include wt-tau or carefully discussing this; likewise, given the depth of analysis one achieves with single cell seq approaches, genetic background issues can be real confounding factors. Was this addressed in the experimental design.

We agree that genetic background is an important potential confounder to consider. The *UAS-tau^R406W^* strain used in this study was iteratively backcrossed to *w1118* for 5 generations as described in our earlier published work (Guo et al., 2018, PMID: 29874575). For comparison, the control strain, *elav-GAL4* / + was generated by outcrossing *elav-GAL4* with the identical *w1118* strain used for backcrossing (*elav-GAL4 / w1118*). In order to clarify these experimental design considerations, our revision includes new explanatory text in the Method. Please also see #4, below, for discussion of tau^WT^.

3) The finding of involvement of the NFkB pathway is interesting, but causality has not been shown. All reviewers thought it would be rather simple to put the idea to test by genetically modulating this pathway and assessing if neuronal loss is rescued.

We thank the reviewer for this suggestion. Given the intriguing association between the Relish (NFkB) regulon and tau-associated cell abundance changes, we tested whether neuron-specific knockdown of *Rel* alters structural brain degeneration in the *elav>tau^R406W^* model. These experiments leveraged 2 independent RNA-interference (RNAi) strains that were previously validated from other published work. Following adult brain histology, we did not detect any significant difference in the vacuolar degeneration caused by tau following pan-neuronal expression of *Relish* RNAi (expression of the *UAS-tau^R406W^* and *UAS-RelRNAi* were both driven by the *elav-GAL4* driver). Our revision includes these new experimental data (Figure 4—figure supplement 8). We have also added new text to our Discussion; however, we interpret these negative data cautiously, since similar genetic manipulations of *Relish* have previously been demonstrated to modulate neurodegeneration in other *Drosophila* neurodegenerative models (Cao et al., 2013, PMID: 23613578; Petersen et al., 2013, PMID:23502677; Kounatidis et al., 2017, PMID:28445733). Moreover, recent studies in mouse models of neurodegeneration suggest that NFkB signaling in neurons may be required for the cell non-autonomous recruitment and activation of microglia (Welch et al., 2022, PMID: 36170369). Thus, while evidence from human genetics strongly supports a causal role for immune processes in AD risk and pathogenesis, further experimentation will be required to definitively resolve cell-type specific mechanisms and contribution of NFkB/Relish in tau-mediated neurodegeneration.

4) the last comment by reviewer 2 was also deemed important. The comparison between species and of an FTD-Tau mutation with AD needs to be toned down.

For the analyses presented in the final section of results and in Figure 5, we had 2 main goals. First, we wanted to confirm the degree of overlap for cell-type specific transcriptional signatures between the *Drosophila* and human brain, providing a useful map for reciprocal forward and reverse translation from single-cell data generated across species. Cross-species overlap between brain cell types is already well established—many studies have noted correspondences between gene expression markers of many excitatory and inhibitory neuronal subtypes, including cholinergic, dopaminergic, and glutamergic cells. Homologies between *Drosophila* and mammalian glial subtypes are also well-established (Freeman, 2015, PMID:25722465, Yildrim et al., 2019; PMID:30443934). Notably, the overlaps shown in Figure 5A (and independently replicated in Figure 5—figure supplement 1B) are not specific to the AD / tau model context; these are broadly generalizable for cross-species interpretation of many other single cell profiling studies of the brain, which we believe that Reviewer 3 appreciated as a strength of the study. In order to further highlight this and help clear up any ambiguity, we repeated our analysis, but excluding human brains from AD cases from Mathys et al. (controls only) and considering overlap with an independent *Drosophila* single-cell dataset comprised of wildtype controls (Davies et al. 2018). The resulting heatmap is consistent with our findings in Figure 5A, revealing cross-species correspondences between many neuron and glial subtypes. These new data are included in our revision (Figure 5—figure supplement 2).

Our second goal was to examine whether NFkB pathway genes are expressed in human neurons and potentially dysregulated in AD, similar to our findings in *elav>tau^R406W^* flies. Our results in Figure 5B-C thus provide support for translation of our findings. The reviewer raises concerns about the validity of this comparison due to differences between our fly tauopathy model and AD, including the (i) reliance on a mutant form of *MAPT* associated with a distinct disease (frontotemporal dementia) and (ii) also the lack of amyloidbeta pathology. In response, we begin by noting that for many decades, similar FTD mutant *MAPT* transgenic mice have been widely used in AD research (e.g., the rTg4510 and PS19 strains which harbor P301L/S variants). Further, while β-amyloid pathology appears to be an important trigger for innate immunity gene expression signatures, several recent studies also highlight an important role for tau (Lee et al., 2021, Wang et al., 2022, Chen et al., 2023).

Regardless, we have performed some new analyses to further address the reviewer’s critiques. First, as suggested, we leveraged a published analysis of scRNAseq “pseudobulk” data from a *MAPT^P301L^* mouse of tauopathy (Lee et al., 2021, PMID:34965428) to evaluate cell-type specific expression of the Rel/NFkB regulon. This analysis demonstrates broadly consistent results with that from human AD brain, including increased NFkB pathway expression in excitatory neurons and microglia (Figure 5—figure supplement 3). Second, in our prior published work on bulk RNAseq (Mangleburg et al., 2020, PMID:32993812), we found an approximately 70% overlap between differentially expressed genes in *elav>tau^WT^* vs. *elav>tau^R406W^*. In order to further examine the conservation between the transcriptional signature induced by wildtype and mutant tau, we plotted longitudinal normalized expression for the innate immune module in each case (vs. control). The results highlight overall similar Tau-triggered increases in innate immunity, with *elav>tau^R406W^* causing a more severe and accelerated gene expression signature (Figure 4—figure supplement 5). Our result is consistent with a notable in vivo study by Bardai et al. (*J Neurosci* 2018, PMID: 29138281) in which 5 different FTDP17 mutant forms of MAPT, including R406W, were compared with wildtype MAPT in *Drosophila* models. The results strongly suggest that R406W (and several other protein coding mutations examined) increase MAPT phosphorylation but share conserved downstream mechanisms leading to neurodegeneration, likely including NFkB signaling.

Besides incorporating these new data (above), we have also carefully reviewed this section of the manuscript for clarity, making a number of additional textual revisions to present a more balanced and cautious interpretation. In the discussion, we also add new tex on the (i) potential limitations for drawing conclusions about AD from models lacking β-amyloid pathology and (ii) the evidence supporting common mechanisms for FTDP17 mutant and wildtype MAPT. We also note the caveat of cross-species comparisons given the uncertain conservation of glia, especially microglia, between mammals and flies.

5) The other issues can likely be addressed by textual changes or added discussion.

Please see additional responses and edits below.

Reviewer #1 (Recommendations for the authors):I have these points that would improve the paper:– Can the authors test whether the neuronal loss in their model is due neurodegeneration rather than developmental toxicity to tau.

Please see our detailed response to Essential revision #1. Briefly, while the changes in cell abundance that we observe at either 1 or 20 days are highly suggestive, except for day 10 where we have replicate samples, our experimental design does not permit cross-sectional robust, statistical analysis, nor can we quantitatively examine changes over time (e.g. 1 vs. 10 days or 1 vs. 20 days). For this reason, our primary analyses highlighting significant cell abundance changes in tau vs. control (Figure 2A) relied on pooled data from all 3 time points, and we also present a cross-sectional replication analysis at day 10. However, our experimental design *does* permit robust analysis of age-dependent changes in gene expression that provide independent, albeit indirect, support for neurodegenerative pathophysiology. In order to address the reviewer request, our revision includes new cross-sectional analyses of cell-specific differential expression changes (Figure 3–Source Data 5). In particular, we highlight neuronal and glial cell types in which tau-induced Relish regulon activity is amplified (or attenuated) with aging (Figure 4—figure supplement 5). We have also added text to provide a more balanced and nuanced discussion of these issues and emphasizing the challenge to disentangle developmental toxicity from age-dependent neurodegeneration.

– Given that they find the NFkB pathway to be involved in tauopathy in a model organism, it would be fascinating if they put this idea to test and show causality by genetically modulating this pathway and rescuing neuronal loss.

As noted in response to Essential revision #3, we have performed new experiments to directly test the hypothesis that neuronal immune pathways are causally linked to tau-mediated neurodegeneration; however, the results were negative. These data are included in the revision and we also carefully discuss published work from other fly models of aging and neurodegeneration as well as mouse tauopathy that strongly suggest NFkB can directly modulate neurodegeneration.

– Can the authors please mention the genetic background of all lines used in the Methods. Where UAS-tau and the wild-type fly that was crossed to elav-Gal4 to serve as a control in the same genetic background?

Please see response to Essential revision #2. We have added the relevant explanatory text to the Methods, and indeed, the control (*elav-GAL4*) strain was outcrossed to same *w1118* strain that we previously used for outcrossing of *UAS-tau^R406W^*.

– The authors should include in the counting step whatever 3' UTR the tau transformation vector had that was used for generating the fly model, since the fast majority of reads should map there rather than in the tau sequence. For now, it seems that NM_016834.5:151-1302 (Methods) represents the CDS.

As detailed above in response to the related comment in the public review, we have repeated the analysis of *MAPT* expression including the UTR sequence, as requested, and updated Figure 3—figure supplement 4. The overall results and interpretation are not substantially changed.

– Figure S6B and S10D are missing a quantification. In addition, for 10D it would be helpful to add a negative control to see how specific the signal is, such as a control fly without the endogenous GFP tag.

We have performed the requested quantifications. Our new results in Figure 2—figure supplement 3C confirm that the signal intensity for multiple markers (DAPI, phalloidin, and repo) are all significantly increased in the *elav>tau^R406W^* adult brain. We have also performed quantification for Figure S10D (now Figure 4—figure supplement 2D in the revision), demonstrating that 78% of neurons and 51% of glia costain for Relish. In order to address specificity, we co-stained flies harboring the GFP-tagged Relish protein with both anti-GFP and anti-Rel. The observed colocalization supports specificity of the Rel-GFP strain that we used to confirm Rel expression in neurons (Figure 4—figure supplement 6).

– It is interesting that glia seem to react strongly to tau. However, it is not clear whether this is cell autonomous – because they also express tau – or as a reaction of the neuronal tau expression. Is the promoter they use neuronal and would we expect any expression in glia? Can they maybe add a panel to Figure S8 with a boxplot showing tau expression levels in glia cell types and neurons.

The *elav-GAL4* driver line used for *MAPT* expression in our model is a well-established and widely used pan-neuronal driver. As requested, we have generated a plot highlighting the specificity of *MAPT* neuronal expression, and we have included this in our revision (Figure 3—figure supplement 4D). Indeed, while we favor a model in which the glial reaction is cell non-autonomous, we cannot completely exclude either a

transient or low level of *MAPT* expression in glia that may contribute in part via an alternative (or additional) cell autonomous mechanism. Indeed, while not a common pathology of AD, other tauopathies like PSP have prominent glial tau aggregates (e.g., tufted astrocytes), and glial tau toxicity has also been modeled in *Drosophila* using a glial-specific driver (Colodner and Feany 2010, PMID: 11408621). Our revision includes new text in the discussion to draw attention to the need for additional studies to dissect cell autonomous vs. cell non-autonomous mechanisms contributing to both neuronal and glial responses in tauopath.

– It is interesting how the authors find multiple regulons (some with >2x larger coefficients than the Rel regulon) to be associated with the degree of vulnerability. For the curious reader it would be helpful to at least point them out and briefly mention the underlying biology.

We thank the reviewer for this suggestion. We have added new text to the results drawing attention to other noteworthy findings from the analysis and the implicated biology.

– Why were in Figure S7 KEGG pathways only annotated for few cell types. This should be explained in legend or annotated more widely, e.g. in all cell types that are lost.

We had originally only provided annotations on pathway enrichment for selected clusters that were discussed in the text, but we agree that this may be confusing. Based on the feedback from the reviewer and also the question raised below regarding the difference between Figure S7B and S9B (now Figure 3—figure supplements 1 and 2) we have removed these highly selective annotations. Instead, in the Figure legend, we refer to Figure 3–Source Data 2 which includes comprehensive pathway enrichment results.

– In Figure S9B not all cell-types shown. Does this mean that no pathways were found in those, maybe they could add this to the legend? And why are the KEGG terms in Figure S7B for a'/b'-KC different than in S9B?

The renamed Figure 3—figure supplement 1B includes all cell clusters with at least 1 enriched KEGG term; we have added clarification to the figure legend. We also note in the legend that readers may refer to Figure 3–Source Data 2 which includes comprehensive annotations, including other curated pathways in addition to KEGG (e.g., GO, Panther and others). For simplicity, in the figure annotation, we restricted our consideration to KEGG pathways. In Figure 3—figure supplement 2B (previously Figure S7B), the selective annotations with functional enrichment analysis considered a subset of differentially expressed genes that were seen consistently in both scRNAseq and bulk tissue RNAseq data; whereas Figure 3—figure supplement 1B considers all differentially expressed genes from the scRNAseq data. Based on the feedback, we agree that this is confusing; we have removed these highly selective annotations and instead refer to Figure 3—figure supplement 1B and Figure 3–Source Data 2 with comprehensive pathway enrichment results.

– Figure S8 should add whether this is counts or log-scale.

The gene expression data in Figure 3—figure supplement 4 are displayed as normalized gene counts. This has been clarified in the figure legend.

– Figure 4D can the authors add explicitly whether this is control and tau cells pooled?

Control and tau cells are pooled in Figure 4D. We have clarified this in the revised figure legend.

Reviewer #2 (Recommendations for the authors):Wu et al. conducted longitudinal single-nucleus RNA sequencing in a *Drosophila* transgenic line expressing pathogenic tau (Arg406 ->Trp) and control to study presenile degenerative dementia with bitemporal atrophy. Their data is consistent with previous findings on Tau neurotoxicity, which significantly affects excitatory neurons in human brain samples and transgenic mice. Intriguingly, intracellular transgenic Tau induced strong transcriptional signatures, aging-like signatures, and an innate immune response, including the NFKB pathway, in the transgenic animals. This dataset provides a valuable resource for exploring dynamic, age-dependent gene expression changes at a cellular level.

We thank the reviewer for this positive feedback. We also want to clarify that beyond frontotemporal dementia / FTDP17, which results from the *MAPT^R406W^* mutation, we believe that our studies also provide important insights on the cell-specific transcriptional mechanisms of tau-mediated neurodegeneration in Alzheimer’s disease (AD). As noted in our response, FTD mutant *MAPT* transgenic mice have been widely used in AD research for decades (e.g. rTg4510 and PS19 which harbor P301L/S variants). In our prior study of *Drosophila* head bulk RNAseq (Mangleburg et al., 2020), we found an approximately 70% overlap between differentially expressed genes in *elav>Tau^WT^* vs. *elav>Tau^R406W^*. This result is consistent with the systematic in vivo study by Bardai et al. (*J Neurosci* 2018, PMID: 29138281) using *Drosophila* transgenic models in which 5 different FTDP17 mutant forms of MAPT, including R406W, were compared with wildtype. The results of this work strongly suggest that R406W (and several other protein coding mutations included) increase MAPT phosphorylation but share conserved downstream mechanisms with wildtype MAPT toxicity, leading to neurodegeneration.

The authors propose that innate immune signatures may serve as predictors of neuronal subtype vulnerability in tauopathies. However, the observed skewing of cell proportions in day-1 animals necessitates stronger evidence to support this hypothesis, which is currently lacking in the manuscript. The paper is primarily descriptive and lacks mechanistic insights. Furthermore, the identified pathways/genes presented in the paper lack orthogonal validation.

As noted in the response to Essential revisions #1 and #3, we have performed additional analyses and made textual revisions to address questions concerning tau developmental toxicity, and we have also attempted additional experiments to directly test whether NFkB pathways may be causal in neurons, which we include in the revision (Figure 4—figure supplement 8). While we feel that evidence from other publications suggest our findings have causal implications, we have taken an overall cautious approach in our interpretations. For example, our abstract notes only that NFkB signaling maybe a “*marker* for cellular vulnerability”. Our revision includes substantial new text clarifying these issues and the pertinent caveats.

1. About the controls: Authors compared Tau transgenic line (Arg406 ->Trp) with the control (GAL4 expressing animals) but not with wt-tau line. They may potentially lead to misinterpretation. Although Wittmann et al. 2001 noted toxicity when wt-tau is expressed, the toxicity is much less compared to Tau transgenic line. Or would another alternative be to use mutant tau animals lacking aggregation-prone regions?

Our goal was to understand mechanisms of tau-mediated neurodegeneration relevant broadly across tauopathies, including AD. As explained above, prior published work, including our own data, support conserved molecular mechanisms of neurodegeneration for wildtype and mutant forms of *MAPT*. Subsequent to the Wittman et al. study, the same group performed additional analyses in which wildtype and mutant MAPT transgene expression levels were more tightly controlled, and these lines were found to have strongly overlapping mechanisms (Bardai et al., *J Neurosci* 2018, PMID: 29138281). Indeed, FTDP17 mutant forms of MAPT have been integral to studies of tau-mediated neurodegeneration across flies, mice, and cellular models, and such findings have been broadly translatable to AD, albeit with important caveats, similar to any experimental model. Nevertheless, as discussed in the response to Essential revision #4, we have performed some new analyses to address some of the reviewer’s concerns (see response to Reviewer 2, point #10). In particular, we leveraged our previously published bulk RNAseq (Mangleburg et al., 2020) in order to examine the conservation of innate immune transcriptional signatures induced by wildtype and mutant tau (Figure 4—figure supplement 1). The results highlight overall similar Tau-triggered increases in innate immunity, with *elav>tau^R406W^* causing a more severe and accelerated gene expression change as expected.

2. It is striking to see the drastic cell proportion changes on day-1 (figure 2B), which may reflect the deficits in neuronal development. Did authors check the expression of transgene expression levels across neuronal subtypes to make sure the vulnerability is not due to a difference in the Tau transgene expression?

Our originally submitted manuscript included an analysis confirming that the affected cell types and differentially expressed genes do not simply correspond to the spatial pattern of *MAPT* transgene expression in the *Drosophila* brain (Figure 3—figure supplement 4). Based on the feedback from Reviewer 1, our revision includes an updated, improved version of this analysis, and we have additionally included a new plot highlighting the lack of correlation between *MAPT* expression and cell abundance (Figure 3—figure supplement 4B). In addition, in our expanded elastic-net regression model (Figure 4–Source Data 4) MAPT expression was excluded as a significant predictor of cell abundance change. Please also see response to Essential revision #1 for discussion of the concern related to tau developmental toxicity.

3. Figure 2B-D, although authors admit that the difference is likely due to the "increase in glial cell abundance from scRNAseq is likely a consequence of proportional changes in single cell suspensions due to neuronal loss," is there a way to quantitatively assess this? Especially authors know the amount of neuronal loss and increase in the glial cells through scRNAseq.

The analyses presented in Figure 2 are indeed an indirect quantification of the proportional changes in neuronal versus glial numbers. Based on our cell count data from the 10-day old flies with triplicate samples, the proportion of neurons in 10x Chromium libraries were reduced from 90% to 83% in controls vs. *elav>tau^R406W^* flies, respectively*.* We have added these estimates to the Results text. Our conclusion that the observed changes likely reflect neuronal loss is additionally informed by our quantification of our experimental data showing largely preserved glial numbers and reduced overall brain volumes (Figure 2D). Lastly, we include an analysis in which confidence intervals for cell abundance changes were computed using an alternative model in which glia were assumed to be unchanging (Figure 2—figure supplement 3A).

4. Line 204-205, authors claim, "93% of tau-induced differentially expressed genes were also triggered by aging in control flies". However, figure 3A does not reflect the 93% similarity. There are more DEGs in age-specific conditions compared to the Tau. The same holds true for the Figure 2B.

The 93% figure is based on consideration of the overall count of unique differentially expressed genes (DEGs), pooling across all cell types. Of the 5,280 tau-triggered DEGs, there is a 93% overlap with the 5,998 aging-induced gene set. However, the degree of overlap varies when considered separately in each cell type, with a range of 0-75%. Indeed, we want to draw attention to the striking difference between overall brain vs. cell type-specific transcriptome overlaps between tau and aging, as well as the distribution of these changes in the adult fly brain. Our revision includes edits to clarify the 93% figure, and we also present the range of cell-specific overlap in Results.

5. Figure 2B, the number of DEG in cluster Lai and Kenyon cells is highly skewed in the Tau transgenic lines. I find it is intriguing to see high number of DEGs in the cells that are degenerating. Since these plots don't tell much about whether they are up or down, it would be good to mention what proportion of the genes are up and down.

Figure 3—figure supplement 1A displays the number of up versus down tau DEGs for each cell cluster. The number of up- vs. down-regulated DEGs for Lai and Kenyon cells are roughly equal, and we have added a mention of this result in the text.

6. The authors claim that among non-neuronal cell types, ensheathing glia, cortex glia, astrocyte-like glia, and hemocytes have the highest number of tau-driven DEGs, but this is not clear from the UMAP in Figure 3C. Additionally, Figure 3C lacks a scale bar, making it difficult to interpret and compare the figure with Figure 3B.

The visualization provided in Figure 3C is intended to complement the plots in panels A and B, highlighting the striking difference between tau- and age- induced cell-type-specific DEGs. Because of the dynamic range and quantitatively stronger impact of aging, the applied scaling tends to minimize the tau-triggered DEGs. In order to address this, we have regenerated Figure 3C without the prior scaling, which better highlights transcriptional changes in the ensheathing glia, and we also have added a label to the scale bar. We have also generated a new dedicated plot of tau DEGs, based on the Figure 3B data with its own, independent scaling which we include as Figure 3—figure supplement 1C.

7. In line 249, the authors claim 90% concordance with previously published datasets, but the data representing this is missing in the paper. Additionally, performing DEG with pseudo-bulk from different clusters and performing DEG to find the concordance may not be very informative. For example, did the authors find consistent gene signatures per cluster when compared with previous datasets? This data should be provided.

We apologize for this misunderstanding. This replication analysis (6 total scRNAseq libraries), including triplicate samples collected each from 10-day-old controls (*elav-GAL4*) and *elav>tau^R406W^*, is in fact a completely new, unpublished scRNAseq dataset that was generated as part of this study. The replication dataset is described in the Methods and we have included associated analytics in the supplemental information (Figure 2—figure supplement 1 and Figure 3–Source Data 3,4); all data has also been deposited in Synapse so that it is available for the community. This dataset permitted our replication analysis for cell abundance changes discussed in the response to Essential revision #1 as well as the replication of cell-type specific differentially expressed genes. The 90% figure refers to the overlap between 3,937 and 4,957 total unique tau-triggered differentially expressed genes at 10-days between the discovery and replication dataset, respectively. In order to examine cell-specific overlaps, we newly considered the full discovery dataset (results from primary age-adjusted analysis, including days 1, 10 and 20. Two-thirds of clusters (61 out of 90)) show a significant overlap with our 10-day-old replication dataset, including excitatory neuron and glial subtypes that are the major focus of our manuscript. We have added these new replication analyses to the Results (Figure 3–Source Data 4).

8. Authors have created an excellent data resource, and it would be interesting to explore the resilience of inhibitory neurons or the vulnerability of excitatory neurons to gain more insights into the cell-type-specific vulnerability or resilience mechanisms. The authors should present a couple of volcano plots showing the differentially expressed genes between important clusters, such as LAI, Kenyon cells, ensheathing glia, etc.

We thank the reviewer for this suggestion. Our revision includes new volcano plots in Figure 3—figure supplements 3 and 5, highlighting selected cell clusters of interest that we discuss in the manuscript, including vulnerable excitatory neurons (e.g., Kenyon cells, Dm3a/b, and Lawf1) and glial subtypes (e.g., ensheathing and perineurial glia and astrocyte-like cells).

9. It would be beneficial for the authors to explore these pathways in greater depth and perform further experimental validation to strengthen the findings using orthogonal approaches. For instance, a rescue experiment where NFKB/Relish is knocked out to see if this modifies Tau toxicity.

Within the scope of this manuscript, we have focused our experimental validation on Relish / NFkB immune pathway. This includes confirming Relish expression in neurons, and we have now added new experimental data directly testing whether genetic manipulation of Relish modulates tau-mediated neurodegeneration. As detailed in the response to Essential revision #3, this result is negative but we have incorporated these data in our revised manuscript along with new discussion. As requested by Reviewer 1, we have also added new text to the discussion highlighting additional pathways of interest based on the results of our elastic net regression analysis.

10. In Figure 5, the authors compared cell-type-specific transcriptional signatures between human Alzheimer's disease (AD) and *Drosophila*. However, some readers may find this comparison difficult to comprehend as the two species differ vastly. Moreover, the Tau mutation that the authors investigated is not associated with AD. Also, in AD, amyloid pathology significantly drives gene expression in immune cells, which is absent in Drosophila. Authors should consider taking the relevant dataset derived from the Arg406 ->Trp patients or from the iPSC-derived cells to validate the observations.

As detailed in our response to Essential revision #4, we have revised the Results and Discussion text to clarify the goals and present a more balanced and cautious interpretation of these cross-species analyses. Also, as noted above, our revision includes new text to justify the use of *elav>tau^R406W^* model for insights relevant to AD, noting the potential caveats, including lack of amyloid-β pathology. We also performed new analyses, as suggested. First, to address the reviewer’s concerns, we repeated the analysis presented in Figure 5A, but excluding human brains from AD cases from Mathys et al. (controls only) and considering overlap with an independent *Drosophila* single-cell dataset comprised of wildtype controls (Davies et al. 2018). The resulting heatmap is consistent with our findings in Figure 5A, revealing cross-species correspondences between many neuron and glial subtypes independent of AD pathology and disease models (Figure 5—figure supplement 2). Second, we leveraged a published analysis of scRNAseq data from a *MAPT^P301L^* mouse of tauopathy (Lee et al., 2021, PMID:34965428) to evaluate cell-type specific expression of the Rel/NFkB regulon in the mammalian brain. This analysis demonstrates broadly consistent results with that from human AD brain, including increased NFkB pathway expression in excitatory neurons and microglia (Figure 5—figure supplement 3). Lastly, we also present new analyses from our prior published work on bulk RNAseq (Mangleburg et al., 2020), where we found an approximately 70% overlap between differentially expressed genes in *elav>tau^WT^* vs. *elav>tau^R406W^*. We therefore plotted longitudinal normalized expression for the innate immune module in each case (vs. control flies), highlighting the conservation of transcriptional signatures induced by wildtype and mutant tau (Figure 4—figure supplement 1). The results highlight overall similar tau-triggered increases in innate immunity, with *elav>tau^R406W^* causing a more severe and accelerated gene expression signature. This result is consistent with Bardai et al. (*J Neurosci* 2018, PMID: 29138281) in which R406W mutant and wildtype *MAPT* were shown to share conserved downstream mechanisms leading to neurodegeneration, likely including NFkB immune pathways.

Reviewer #3 (Recommendations for the authors):As mentioned above, I feel this is an elegant and nicely described study. It provides a further example of the power of single-cell transcriptomic to assess disease genes, showing that in such a fashion one can assess the impact of cell-types, but also the actual genes that are altered. It also shows that the findings can be transferred to patient tissue, thus integrating previous published human sc-data.There are a few points that I feel might be important to take into account.– The authors show that ratios of cells are changed in response to tau-GOF, however no explanation is given. What is the basis of this alteration? Cell-death, changes of differentiation/proliferation (it seems the GOF is throughout development in "elav" cells).

As detailed in response to Essential revision #1, we believe that the changes in cell proportion are likely due to a reduction in neuronal numbers, reflecting a combination of developmental toxicity and aging-dependent neurodegeneration. Based on our experimental studies and other published evidence, we believe that glial proliferation is unlikely a major contributor (if at all present). Our revision includes new text in the discussion providing a more balanced and nuanced discussion of these issues, and emphasizing the challenge to disentangle developmental toxicity from age-dependent neurodegeneration.

– The integration and comparison of fly/human data is a bit short, I could not follow the process of how this was achieved, what framework the authors used etc.

We regret that this was unclear. As detailed in response to Essential revision #4, we have tried to clarify with new text in the Results, Discussion and Figure 5 legend, as well as the addition of new analytic results to address concerns raised primarily by Reviewer 2.

– Conceptually, I think it is nice to see the in-silico analysis of the tau-GOF and aging, however I feel some in vivo validation might have been beneficial to support the claims of affected cell-types and differential expressed genes.While I fully agree that an extensive genetic analysis may be beyond the scope of the current paper, a proof-of-concept analysis would have been supportive of the validity of the data.

Our original manuscript included experimental validation demonstrating that (1) the apparent increases in glial cell abundance is likely due to changes in cell proportions, and we also (2) confirmed the expression of Relish in both neurons and glia of the adult fly brain. For our revision, we were guided by the requested Essential Revision #3 (see detailed response), and we have therefore performed additional experiments directly testing whether manipulation of Relish/NFkB in neurons alters tau-induced neurodegeneration. While the results of these experiments were negative, we have incorporated them into the results and discussion, along with discussion of other published studies and potential future work that might further support a causal role for NFkB immune pathways in tauopathy.